# INTERPRETABILITY ILLUSIONS IN THE GENERALIZATION OF SIMPLIFIED MODELS

## ABSTRACT

A common method to study deep learning systems is to simplify model representations—for example, using singular value decomposition to visualize the model's hidden states in a lower dimensional space. This approach assumes that the results of these simplifications are faithful to the original model. Here, we illustrate an important caveat to this assumption: even if the simplified representations can accurately approximate the full model on the training set, they may fail to accurately capture the model's behavior out of distribution; the understanding developed from simplified representations may be an illusion. We illustrate this by training Transformer models on controlled datasets with systematic generalization splits, focusing on the Dyck balanced-parenthesis languages. We simplify these models using tools like dimensionality-reduction and clustering, and find clear patterns in the resulting representations. We then explicitly test how these simplified proxies match the original models behavior on various out-of-distribution test sets. Generally, the simplified proxies are less faithful out of distribution. For example, in cases where the original model generalizes to novel structures or deeper depths, the simplified versions may fail to generalize, or may generalize too well. Even model simplifications that do not directly use data can be less faithful out of distribution. Finally, we show the generality of our results by extending to a more naturalistic language modeling task on computer code, and show similar gaps between the original model and the simplified proxies. Our experiments raise questions about the extent to which mechanistic interpretations derived using tools like SVD can reliably predict what a model will do in novel situations.

## 1 INTRODUCTION

How can we understand deep learning models? Often, we begin by simplifying the model, or its representations, using tools like dimensionality reduction, clustering, and discretization. We then interpret the results of these simplifications—for example finding dimensions in the principal components that encode a task-relevant feature. In other words, we are essentially replacing the original model with a simplified proxy model which uses a more limited—and thus easier to interpret—set of features. By analyzing these simplified proxy models, we hope to understand at an abstract level how the system solves a task. Ideally, this understanding could help us to predict how the model will behave in unfamiliar situations, and thereby to anticipate failure cases and audit for potentially unsafe behavior.

However, in order to arrive at understanding by simplifying a model, we have to assume that the result of the simplification is a relatively *faithful* proxy for the original model. For example, we need to assume that the principal components of the model representations, by capturing most of the variance, are thereby capturing the details of the model's representations that are relevant to its computations. This assumption may not be valid. First, some model simplifications, like PCA, are not computed solely from the model itself; they are calculated with respect to the model's representations for a particular collection of inputs, and therefore depend on the input data distribution. Second, even when a simplification does not explicitly depend on the training distribution, it might appear faithful on in-distribution evaluations, but fail to capture the model's behavior over other distributions. Thus, it is important to understand the extent to which simplified proxy models characterize the behavior of the underlying model beyond this restricted data distribution.

In this work, we therefore study how models, and their simplified proxies, generalize out of distribution. We focus on small-scale Transformer (Vaswani et al., 2017) language models. We first consider models trained on the Dyck balanced-parenthesis languages. These languages have been studied in prior work on characterizing the computational expressivity of Transformers (e.g., Hewitt et al., 2020; Ebrahimi et al., 2020; Yao et al., 2021; Weiss et al., 2021; Murty et al., 2023; Wen et al., 2023), and admit a variety of systematic generalization splits, including generalization to unseen structures, different sequence lengths, and deeper hierarchical nesting depths. First, we simplify and analyze the model's representations, e.g. by visualizing their first few singular vectors. This analysis reveals meaningful structure, illustrating how model simplification can appear to produce a working understanding of the algorithm that the underlying model implements. Next, for each simplification, we explicitly construct the corresponding simplified proxy models—for example, replacing the model's key and query representations with their projection onto the top-k singular vectors—and evaluate how the original models, and their simpler proxies, generalize to out-of-distribution test sets.

We find that the simplified proxy models are not as faithful to the original models out of distribution. While the proxies capture model attention computations and behavior fairly accurately on the training data, they reveal unexpected *generalization gaps* on out-of-distribution tests—the simplified model often *underestimates* the generalization performance of the original model, contrary to intuitions from classic generalization theory (Valiant, 1984; Bartlett & Mendelson, 2002), and recent explanations of grokking (Merrill et al., 2023). However, under certain data-independent simplifications the simpler model actually outperforms the original model; once again, this indicates a mismatch between the original model and its simplification. We elucidate these results by identifying some features of the model's representations that the simplified proxies are capturing and missing, and how these relate to human-written transformer algorithms for this task (Yao et al., 2021). We show that different simplifications produce different kinds of divergences from the original model. Finally, we show that our results extend to other datasets, including a more naturalistic language modeling task on computer code, with distribution shifts to other programming languages.

Our results raise a key question for understanding deep learning models: If we simplify a model in order to interpret it, will we still accurately capture model computations and behaviors out of distribution? We reflect on this issue; the related challenges in fields like neuroscience; and some broader question about the relationship between complexity and generalization, and the relationship between different levels of analysis of a system, from representation to algorithms and principles.

## 2 SETTING

### 2.1 DYCK LANGUAGES

Dyck-$k$ is the family of balanced parenthesis languages with up to $k$ bracket types. Following the notation of Wen et al. (2023), the vocabulary of Dyck-$k$ is the words $\{1, \ldots, 2k\}$, where, for any $t \in [k]$, the words $2t - 1$ and $2t$ are the opening and closing brackets of type $t$, respectively. Given a sentence $w_1, \ldots, w_n$, the nesting depth at any position $i$ is defined as the difference between the number of opening brackets in $w_{1:i}$ and the number of closing brackets in $w_{1:i}$. As in prior work (Yao et al., 2021; Murty et al., 2023; Wen et al., 2023), we focus on bounded-depth Dyck languages (Hewitt et al., 2020), denoted Dyck-$(k, m)$, where $m$ is the maximum nesting depth.

We focus on Dyck for two main reasons. On one hand, the Dyck languages exhibit several fundamental properties of both natural and programming languages—namely, recursive, hierarchical structure, which gives rise to long-distance dependencies. For this reason, Dyck languages have been widely studied in prior work on the expressivity of Transformer language models (Hewitt et al., 2020; Yao et al., 2021), and in interpretability (Wen et al., 2023). On the other hand, these languages are simple enough to admit simple, human-interpretable algorithms (see Section 2.3).

**Generalization splits** For our main analysis, we train models on Dyck-$(20, 10)$, the language with 20 bracket types and a maximum depth of 10, following Murty et al. (2023). To create generalization splits, we follow Murty et al. (2023) and start by sampling a training set with 200k training sentences—using the distribution described by Hewitt et al. (2020)—and then generate test sets with respect to this training set. Next, we recreate the structural generalization split described by Murty et al. (2023) by sampling sentences and discarding seen sentences and sentences with *unseen* bracket structures (**Seen struct**). The bracket structure of a sentence is defined as the sequence of opening

and closing brackets (e.g., the structure of `([])[]` is `OOCCOC`). The above sampling procedure results in a shift in the distribution of sentence lengths, with the *Seen struct* set containing much shorter sentences than the training set.[1] Therefore, we create two equal-sized structural generalization splits, **Unseen struct (len $\leq$ 32)** and **Unseen struct (len > 32)**, by sampling sentences, discarding sentences with seen structures, and partitioning by length. Finally, we create a **Unseen depth** generalization set by sampling sentences from Dyck-$(20, 20)$ and only keeping those sentences with a maximum nesting depth of at least 10. All generalization sets have 20k sentences. More details are provided in Appendix A.1. For example, different generalization splits are illustrated in Appendix Tab. 1, and the distribution of sentence lengths is plotted in Appendix Figure 6.

**Evaluation** Following Murty et al. (2023), we evaluate models' accuracy at predicting closing brackets that are at least 10 positions away from the corresponding opening bracket, and score the prediction by the closing bracket to which the model assigns the highest likelihood.

## 2.2 Transformer Language Models

The Transformer (Vaswani et al., 2017) is a neural network architecture for processing sequence data. The input is a sequence of tokens $w_1, \ldots, w_N \in \mathcal{V}$ in a discrete vocabulary $\mathcal{V}$. At the input layer, the model maps the tokens to a sequence of $d$-dimensional embeddings $\boldsymbol{X}^{(0)} \in \mathbb{R}^{N \times d}$, which is the sum of a learned token embedding and a positional embedding. Each subsequent layer $i$ consists of a multi-head attention layer (MHA) followed by a multilayer perceptron layer (MLP): $\boldsymbol{X}^{(i)} = \boldsymbol{X}^{(i-1)} + \text{MHA}^{(i)}(\boldsymbol{X}^{(i-1)}) + \text{MLP}^{(i)}(\boldsymbol{X}^{(i-1)} + \text{MHA}^{(i)}(\boldsymbol{X}^{(i-1)}))$.[2] Following the presentation of Elhage et al. (2021), multi-head attention (with $H$ heads) can be written as

$$\text{MHA}(\boldsymbol{X}) = \sum_{h=1}^{H} \text{softmax}(\boldsymbol{X}\boldsymbol{W}_Q^h(\boldsymbol{X}\boldsymbol{W}_K^h)^\top)\boldsymbol{X}\boldsymbol{W}_V^h\boldsymbol{W}_O^h,$$

where $\boldsymbol{W}_Q^h, \boldsymbol{W}_K^h, \boldsymbol{W}_V^h \in \mathbb{R}^{d \times d_h}$ are referred to as the *query*, *key*, and *value* projections respectively, and $\boldsymbol{W}_O^h \in \mathbb{R}^{d_h \times d}$ projects the output value back to the model dimension. The MLP layer operates at each position independently; we use a two-layer feedforward network: $\text{MLP}(\boldsymbol{X}) = \sigma(\boldsymbol{X}\boldsymbol{W}_1)\boldsymbol{W}_2$, where $\boldsymbol{W}_1 \in \mathbb{R}^{d \times d_m}, \boldsymbol{W}_2 \in \mathbb{R}^{d_m \times d}$, and $\sigma$ is the ReLU function. The output of the model is a sequence of token embeddings, $\boldsymbol{X}^{(L)} \in \mathbb{R}^{N \times d}$. We focus on autoregressive Transformer language models, which define a distribution over next words, given a prefix $w_1, \ldots, w_{i-1} \in \mathcal{V}$: $p(w_i \mid w_1, \ldots, w_{i-1}) \propto \exp(\theta_{w_i}^\top \boldsymbol{X}_{i-1}^{(L)})$, where $\theta_{w_i} \in \mathbb{R}^d$ is a vector of output weights for word $w_i$.

## 2.3 Transformer Algorithms for Dyck

Our investigations will be guided by a human-written algorithm for modeling Dyck languages with Transformers (Yao et al., 2021), which we review here. The construction uses a two-layer autoregressive Transformer with positional encodings and one attention head per layer.

**First attention layer: Calculate bracket depth.** The first attention layer calculates the bracket depth at each position, defined as the number of opening brackets minus the number of closing brackets. One way to accomplish this is using an attention head that attends uniformly to all tokens and uses one-dimensional value embeddings, with opening brackets having a positive value and closing brackets having a negative value. At position $t$, the attention output will be $\text{depth}(w_{1:t})/t$.

**First MLP: Embed depths.** The output of the first attention layer is a scalar value encoding depth. The first-layer MLP maps these values to orthogonal depth embeddings, which can be used as keys and queries in the next attention layer.

**Second attention layer: Bracket matching.** The second attention layer uses depth embeddings to find the most recent unmatched opening bracket. At position $i$, the most recent unmatched opening bracket is the bracket at position $j$, where $j \leq i$ is the highest value such that $w_j$ is an opening bracket and $\text{depth}(w_{1:j}) = \text{depth}(w_{1:i})$.

---

[1]The longest sentence in our *Seen struct* set has a length of 30.
[2]The standard Transformer also includes a layer-normalization layer (Ba et al., 2016), which we omit here.

## 3 APPROACH

**Scope of our analysis**   For Transformer LMs, mechanistic interpretations can be divided into two stages: *circuit identification* and *explaining circuit components*. The first stage involves identifying the subgraph of model components that are involved in some behavior. The second stage involves characterizing the computations of each component. Various automated circuit identification methods have been developed (e.g. Vig et al., 2020; Meng et al., 2022; Conmy et al., 2023), but relatively less work has focused on the second stage of interpretation, which we focus on here. We use small models with one attention head per layer, where the "circuit" is largely unambiguous, and aim to characterize the algorithmic role of each component, focusing in particular on attention heads.

**Interpretation by model simplification**   Our main focus is on understanding the mechanism by which the model identifies matching brackets, so we focus here on the keys and queries for the second-layer attention head. We evaluate two data-dependent methods of simplifying keys and queries, and a third data-agnostic method that simplifies the resulting attention pattern:

*Simplifying key and query embeddings.* Our first two approaches aim to characterize the attention mechanism by examining simpler representations of the key and query embeddings. To do this, we collect the embeddings for a sample of 1,000 training sequences. **Dimension reduction:** We calculate the singular value decomposition of the concatenation of key and query embeddings. For evaluation, we project all key and query embeddings onto the first $k$ singular vectors before calculating the attention pattern.This approach is common in prior work in mechanistic interpretability (e.g. Lieberum et al., 2023). **Clustering:** We run k-means on the embeddings, clustering keys and queries separately. For evaluation, we replace each key and query with the closest cluster center prior to calculating the attention pattern. This approach has precedent in a long line of existing work on extracting discrete rules from RNNs (e.g. Omlin & Giles, 1996; Jacobsson, 2005; Weiss et al., 2018; Merrill & Tsilivis, 2022), and can allow us to characterize attention using discrete case analysis.

*Simplifying the attention pattern.* The Transformer algorithm for Dyck described above (§2.3) uses hard attention for the second layer (i.e., the attention head attends to the matching bracket and predicts whichever bracket receives the highest score). While our trained model uses standard softmax attention, this suggests a simplification where the soft attention is replaced with **one-hot attention** to highest-scoring key. This simplification has the advantage of being data-agnostic; it is purely a change to the model. We evaluate this simplification by replacing the computed attention pattern with one-hot attention, with all attention assigned to the highest-scoring key.

## 4 CASE STUDY: DYCK LANGUAGE MODELING

In this section, we train two-layer Transformer language models on Dyck languages. In Appendix B.1, we illustrate how we can attempt to reverse-engineer the algorithms these model learn by inspecting simplified model representations, using visualization methods that are common in prior work (e.g. Liu et al., 2022; Power et al., 2022; Zhong et al., 2023; Chughtai et al., 2023; Lieberum et al., 2023). In this section, we quantify how well these simplified proxy models predict the behavior of the underlying model. First, we plot approximation quality metrics for different model simplifications and generalization splits (§4.1), finding a consistent generalization gap. Then we try to explain why this generalization gap occurs by analyzing the approximation errors (§4.2). We include additional results in Appendix B, including more analysis of simplified models (B.4 and B.3) and experiments with other datasets (B.6).

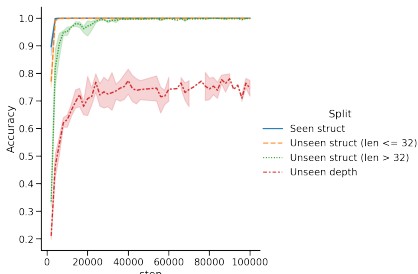

Figure 1: Accuracy at predicting closing brackets on different generalization splits over the course of training (§2.1), averaged over three random seeds.

**Model and training details**   We train two-layer Transformer language models on the Dyck-$(20, 10)$ training data described in the previous section. The model uses learned absolute positional embeddings. Each layer has one attention head, one MLP, and layer normalization, and the model

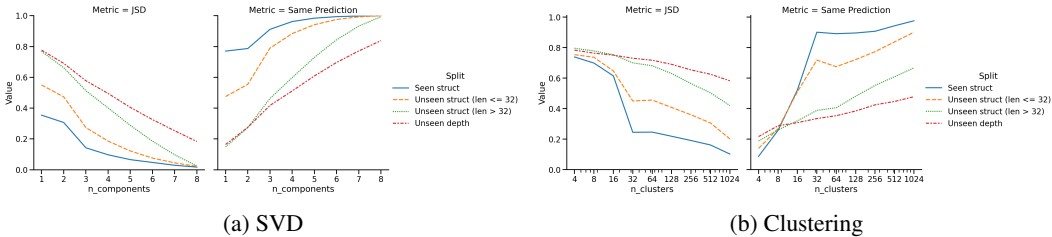

(a) SVD        (b) Clustering

Figure 2: Approximation quality after applying two simplifications to the key and query embeddings, SVD (*left*) and clustering (*right*). *JSD* is the average Jensen-Shannon Divergence between the attention patterns of the original and simplified models, and *Same Prediction* measures whether the two models make the same prediction at the final layer.

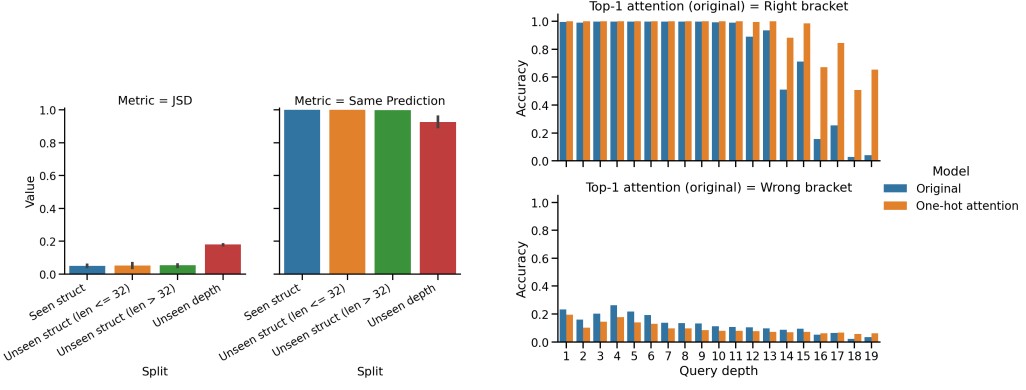

(a) Approximation metrics for one-hot attention.     (b) Accuracy on the depth generalization split.

Figure 3: Approximation quality and accuracy after replacing the second-layer attention pattern with a one-hot attention pattern that assigns all attention to the highest scoring key, averaged over three models trained with different random seeds. One-hot attention is a faithful approximation on all generalization splits except for the depth generalization split (Fig. 3a). This difference illustrates that a simplification which is faithful in some out-of-distribution evaluations may fail in others. In depth generalization, the one-hot attention model slightly out-performs the original model (Fig. 3b)—particularly at higher depths and in cases where the original model attends to the correct location—thereby over-estimating how well the original model will generalize.

has a hidden dimension of 32. Details about the model and training procedure are in Appendix A.2 and A.3. Fig. 1 plots the bracket-matching accuracy over the course of training, averaged over three runs. Consistent with Murty et al. (2023), we find that the models reach perfect accuracy on the in-domain held-out set early in training, and reach near-perfect accuracy on the structural generalization set later. On the depth generalization split, the models achieve approximately 75% accuracy.

## 4.1 GENERALIZATION GAPS

We evaluate approximation quality using two metrics: the Jensen-Shannon Divergence (*JSD*), a measure of how much the attention pattern diverges from the attention pattern of the original model; and whether or not the two models make the *Same Prediction* at the final layer. Fig. 2b shows these metrics after simplifying the key and query representations. The simplified models correspond fairly well to the original model on the in-distribution evaluation set, but there are consistent performance gaps on the generalization splits. This suggests that these simplified methods underestimate generalization. Fig. 3 shows the effect of replacing the attention pattern with one-hot attention. One-hot attention is a faithful approximation on all generalization splits except for the depth generalization split (Fig. 3a). In this setting, the one-hot attention model slightly out-performs the original model (Fig. 3b), in a sense over-estimating how well the model will generalize.

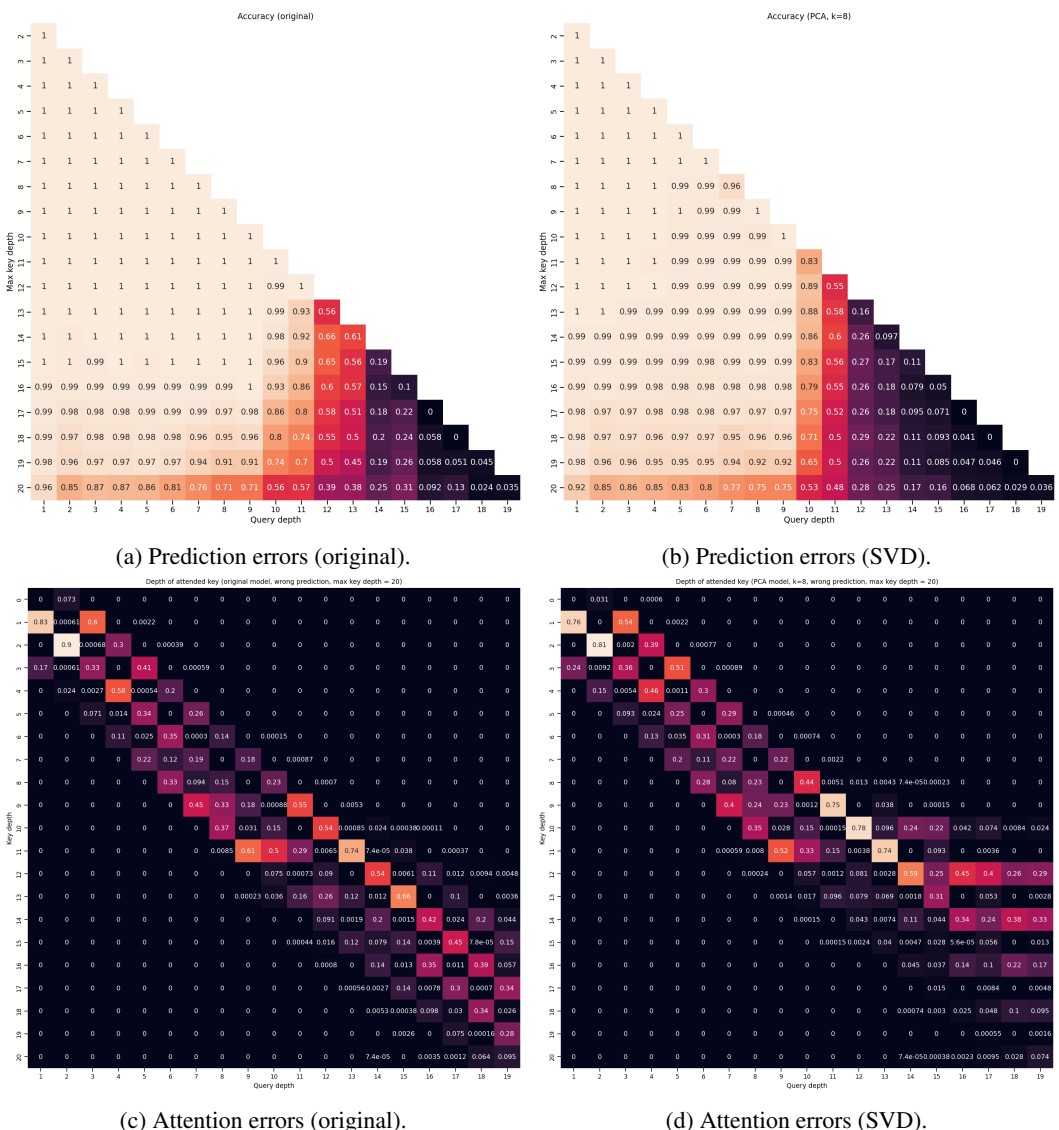

(a) Prediction errors (original).

(b) Prediction errors (SVD).

(c) Attention errors (original).

(d) Attention errors (SVD).

Figure 4: Errors of the original model and a rank-8 SVD simplification on the depth generalization test set. Figures 4a and 4b plot the prediction accuracy, broken down by the depth of the query and the maximum depth among the keys. Figures 4c and 4d plot the depth of the token with the highest attention score, broken down by the true target depth, considering only incorrect predictions.

## 4.2 WHAT DO SIMPLIFIED MODELS MISS?

**Comparing error patterns** Figures 4a and 4b plot the errors made by the original model and a rank-8 SVD approximation, broken down by query depth and the maximum depth in the sequence. While both models have a similar overall error pattern, the rank-8 model somewhat underestimates generalization, performing poorly on certain out-of-domain cases where the original model successfully generalizes (i.e. depths 11 and 12). Figures 4c and 4d plot the errors made by the attention mechanism, showing the depth of the keys receiving the highest attention scores in cases where the final prediction of the original model is incorrect. In this case, both models have similar error patterns on shallower depths, attending to depths either two higher or two lower then the target depth; this error is in line with Fig. 8c. However, the error patterns diverge on depths greater than ten, suggesting that the lower-dimension model can explain why the original model makes mistakes in some in-domain cases, but not out-of-domain.

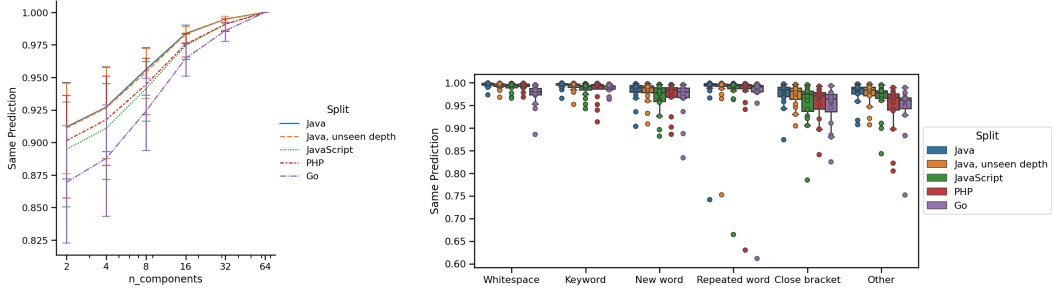

(a) Approximation quality.  (b) Approximation quality by prediction type with 16 components.

Figure 5: Prediction similarity on CodeSearchNet after reducing the dimension of the key and query embeddings using SVD, filtered to the subset of tokens that the original model predicts correctly. We apply dimension reduction to each attention head independently and aggregate the results over attention heads and over models trained with three random seeds. Each model has four layers, four attention heads per-layer, and a head dimension of 64. The prediction similarity between the original and simplified models is consistently higher on in-distribution examples (Java) relative to examples with deeper nesting depths or unseen languages (Fig. 5a). Fig. 5b breaks down the results by prediction type, for a single model using 16 SVD components. Each point on the plot shows the prediction similarity after simplifying one attention head. The gap between in-distribution and out-of-distribution approximation scores is greater on some types of predictions than others, perhaps because these predictions depend more on precise, context-dependent attention.

**One-hot attention** In Fig. 3b, we compare the accuracy of the Dyck model on the depth generalization split before and after replacing the attention pattern with a one-hot simplification. The accuracy is broken down by the nesting depth at the query position, and by whether the original model assigns the highest attention score to the correct opening bracket. In this setting, simplifying the model—by assuming that the top-1 attention score determines the model's prediction—leads to an over-estimate of how well the model will generalize.

## 5 GENERALIZATION GAPS IN MORE NATURALISTIC SETTINGS

In this section, we investigate whether our findings extend to larger models trained on a more practical context: predicting the next character in a dataset of computer code. Code completion has become a common use case for large language models, with applications in developer tools, personal programming assistants, and automated agents (e.g. Anil et al., 2023; OpenAI, 2023). This task is a natural transition point from the Dyck setting, requiring both "algorithmic" reasoning (including bracket matching) and more naturalistic language modeling. Specifically, we train character-level language models on the CodeSearchNet dataset (Husain et al., 2019), which is made up of functions in a variety of programming languages. As in the Dyck languages, we define the bracket nesting depth as difference between the number of opening and closing brackets at each position, treating three pairs of characters as brackets (`()`, `{}`, `[]`). We train models on Java functions with a maximum nesting depth of three and construct two kinds of generalization split: Java functions with deeper nesting depths (**Java, unseen depth**); and functions with a seen depth but written in an unseen language (**JavaScript**, **PHP**, **Go**). See Appendix B.5 for more dataset and training details.

**Generalization gaps: Code completion** We train Transformer language models with four layers, four attention heads per-layer, and an attention embedding size of 64, and train models with three random initializations. We measure the effect of simplifying each attention head independently: for each attention head, we reduce the dimension of the key and query embeddings using SVD and then calculate the percentage of instances in which the original and simplified model make the same prediction. Figure 5a plots the average prediction similarity, filtering to cases where the original model is correct. The prediction similarity is consistently higher on Java examples and lower on unseen programming languages, suggesting that this task also gives rise to a generalization gap. On the other hand, there is no discernible generalization gap on the Java depth generalization split; this could be because in human-written code, unlike in Dyck, bracket types are correlated with other

contextual features, and so local model simplifications may have less of an effect on prediction similarity.

**Comparing subtasks** Which aspects of the code completion task give rise to bigger or smaller generalization gaps? In Figure 5b, we break down the results by the type of character the model is predicting: whether it is a **Whitespace** character; part of a reserved word in Java (**Keyword**); part of a new identifier (**New word**); part of an identifier that appeared earlier in the function (**Repeated word**); a **Close bracket**; or any **Other** character, including opening brackets, semicolons, and operators. We plot the results for each attention head in a single model after projecting the key and query embeddings to the first 16 SVD components and include results for other models and dimensions in Appendix B.5. The results vary depending on the prediction type, including both the overall approximation scores and the relative difference between approximation quality on different subsets. In particular, the generalization gap is larger on subtasks that can be seen as more "algorithmic", including predicting closing brackets and copying identifiers from earlier in the sequence. The gap is smaller when predicting whitespace characters and keywords, perhaps because these predictions rely more on local context.

**Comparing attention heads** In Figure 5b, we can observe that some prediction types are characterized by outlier attention heads: simplifying these attention heads leads to much lower approximation quality, and larger gaps in approximation quality between in-distribution and out-of-distribution samples. This phenomenon is most pronounced in the *Repeated word* category, where simplifying a single attention head reduces the prediction similarity score to 75% on in-domain samples and as low as 61% on samples from unseen languages. In the appendix (B.5), we find evidence that this head implements the "induction head" pattern, which has been found to play a role in the emergence of in-context learning in Transformer language models (Elhage et al., 2021; Olsson et al., 2022). This finding suggests that the low-dimensional approximation underestimates the extent to which the induction head mechanism will generalize to unseen distributions.

# 6 RELATED WORK

**Circuit analysis** Research on reverse-engineering neural networks comes in different flavors, each focusing on varying levels of granularity. A growing body of literature aims to identify Transformer *circuits* (Olsson et al., 2022). Circuits refer to components and corresponding information flow patterns that implement specific functions, such as indirect object identification (Wang et al., 2023), variable binding (Davies et al., 2023), arithmetic operations (e.g., Stolfo et al., 2023; Hanna et al., 2023), or recalling factual associations (e.g., Meng et al., 2022; Geva et al., 2023). More recently, there have been attempts to scale up this process by automating circuit discovery (Conmy et al., 2023; Davies et al., 2023) and establishing hypothesis testing pipelines (Chan et al., 2022; Goldowsky-Dill et al., 2023). In addition to the progress made in circuit discovery, prior research has highlighted some challenges. For example, contrary to earlier findings (Nanda et al., 2022), small models trained on prototypical tasks, such as modular addition, exhibit a variety of qualitatively different algorithms (Zhong et al., 2023; Pearce et al., 2023). Even in setups with strong evidence in favor of particular circuits, such as factual associations, modules that exhibit the highest causal effect in *storing* knowledge may not necessarily be the most effective choice for *editing* the same knowledge (Hase et al., 2023). Our results highlight an additional potential challenge: a circuit identified using one dataset may behave differently out of distribution.

**Analyzing attention heads** Because attention is a key component of the Transformer architecture (Vaswani et al., 2017), it has been a prime focus for analysis. Various interesting patterns have been reported, such as attention heads that particularly attend to separators, syntax, position, or rare words (e.g., Clark et al., 2019; Vig, 2019). These patterns also provide insights into how much damage the removal of the said head would cause to the network (Guan et al., 2020). The extent to which attention can explain the model's predictions is a subject of debate. Some argue against using attention as an explanation for the model's behavior, as attention weights can be manipulated in a way that does not affect the model's predictions but can yield significantly different interpretations (e.g., Pruthi et al., 2020; Jain & Wallace, 2019). Others have proposed tests for scenarios where attention can serve as a valid explanation (Wiegreffe & Pinter, 2019). It is interesting to consider our results on one-hot attention simplification and depth generalization through the lens of this debate.

Top-1 attention accuracy is highly predictive of model success in most cases; however, on deeper structures top-1 attention and accuracy dissociate. Thus, while top attention could be an explanation in most cases, it fails in others. However, our further analyses suggest that this is due to increased attention to other elements, even if the top-1 is correct—thus, attention patterns may still explain these errors, as long as we do not oversimplify attention before interpreting it.

**Neuron-level interpretability**  A more granular approach to interpretability involves examining models at the neuron level to identify the concepts encoded by individual neurons. By finding examples that maximally activate specific "neurons"— a hidden dimension in a particular module like an MLP or in the Transformer's residual stream— one can deduce their functionality. These examples can be sourced from a dataset or generated automatically (Belinkov & Glass, 2019). It has been observed that a single neuron sometimes represents high-level concepts consistently and even responds to interventions accordingly. For example, Bau et al. (2020) demonstrated that by activating or deactivating the neuron encoding the "tree" concept in an image generation model, one can respectively add or remove a tree from an image. Neurons can also work as n-gram detectors or encode position information (Voita et al., 2023). However, it is essential to note that such example-dependent methods could potentially lead to illusory conclusions (Bolukbasi et al., 2021).

**Related challenges in neuroscience**  Neuroscience also relies on stimuli to drive neural responses, and thus similarly risks interpretations that may not generalize. For example, historical research on retinal coding used simple bars or grids as visual stimuli. However, testing naturalistic stimuli produced many new findings (Karamanlis et al., 2022)—e.g., some retinal neurons respond only to an object moving differently than its background (Ölveczky et al., 2003), so their function could never be determined from simple stimuli. Thus, intepretations of retinal computation drawn from simpler stimuli did not fully capture its computations over all possible test distributions. Neuroscience and model interpretability face common challenges from interactions between different levels of analysis (Churchland & Sejnowski, 1988): we wish to understand a system at an abstract algorithmic level, but its actual behavior may depend on low-level details of its representational implementation.

**The relationship between complexity and generalization**  Classical generalization theory suggests that simpler models generalize better (Valiant, 1984; Bartlett & Mendelson, 2002), unless datasets are massive; however, in practice overparameterized deep learning models generalize well (Nakkiran et al., 2021; Dar et al., 2021). Recent theory has explained this via *implicit regularization* effectively simplifying the models (Neyshabur, 2017; Arora et al., 2019), e.g. making them more compressible (Zhou et al., 2018). Our results reflect this complex relationship between model simplicity and generalization. We find that data-dependent approaches to simplifying the models' representations impair generalization. However, a *data-independent* simplification (hard attention) allows the simplified model to generalize better to high depths. This result reflects the match between the Transformer algorithm for Dyck and the inductive bias of hard attention, and therefore echoes some of the classical understanding about model complexity and the bias-variance tradeoff.

## 7 Conclusion

In this work, we simplified Transformer language models using tools like dimensionality reduction to investigate their computations. We then compared how the original models and their simplified proxies generalized out-of-distribution. On one hand, we found that the low-dimensional simplifications of the model's representations revealed meaningful structure, suggesting that these models learn algorithms that resemble human-written constructions. On the other hand, we found that such simplifications fail to predict the behavior of the underlying model under various distribution shifts despite being faithful on in-distribution evaluations. Overall, these results highlight the limitations of interpretability methods that depend upon simplifying a model, and the importance of evaluating model interpretations out of distribution.

**Limitations**  This study focused on small-scale models trained on a limited range of tasks. Across the settings we considered, we found that simplified proxy models give rise to consistent generalization gaps on a variety of distribution shifts — particularly on algorithmic predictions. However, future work will study how these findings apply to larger-scale models trained on other families of tasks, such as large (natural) language models.

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

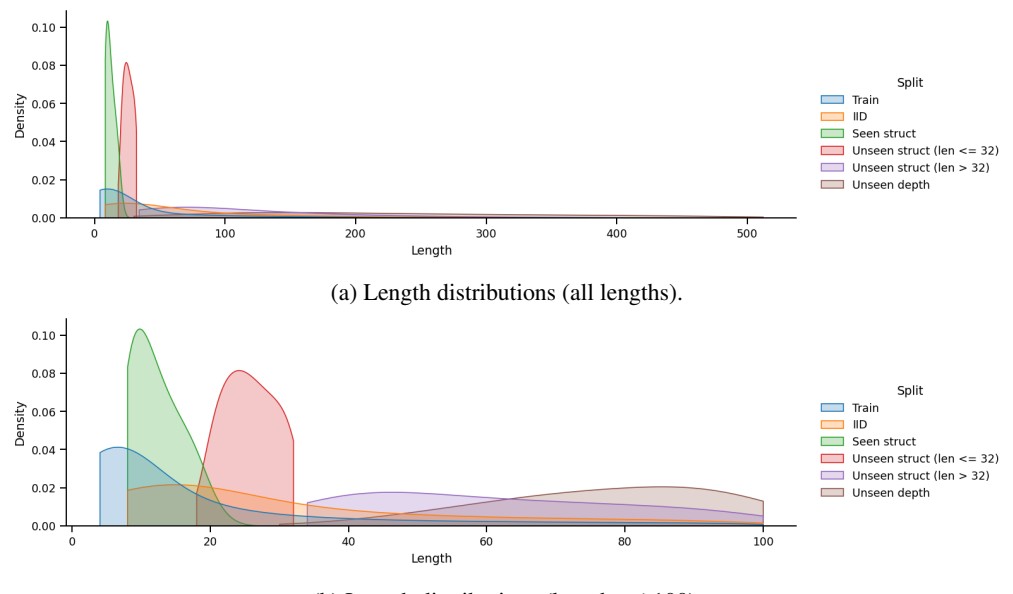

(a) Length distributions (all lengths).

(b) Length distributions (lengths $\leq 100$).

Figure 6: Distribution of sentence lengths in different Dyck structural generalization splits. Figure 6a shows the full distribution and Figure 6b shows the distribution filtered to sequences with length $\leq 100$. These densities are estimated using the Seaborn library kernel density estimation method, with a bandwidth adjustment factor of three. (Note that all sequences of balanced parentheses have even lengths, which is smoothed over in the plots.) For reference, we also include an additional split created by sampling from the same distribution as the training data, but discarding sentences that appeared in the training set (**IID**); see Section A.1 for further discussion.

| Subset | | Illustrative Samples | | | |
|---|---|---|---|---|---|
| *Train* | Random samples from Dyck-(3,2) with three different bracket types of ( ) [ ] { }. All sentences have the maximum nesting depth of two. | Sentence | ( ) { } | [ [ ] ( ) ] | { } ( { } ( ) ) |
| | | Structure | OCOC | OOCOCC | OCOOCOCC |
| | | Depth | 1111 | 122221 | 11122221 |
| *Seen Struct* | Random samples with bracket structures that appeared in the Train set, but with different bracket types. | Sentence | [ ] ( ) | [ ] [ ] | ( ) { [ ] ( ) } |
| | | Structure | OCOC | OOCOCC | OCOOCOCC |
| | | Depth | 1111 | 122221 | 11122221 |
| *Unseen Struct* | Random samples with bracket structures that have not appeared in the Train set, but have the same maximum nesting depth of two. | Sentence | ( ) [ ] { } | { [ ] } ( { } ) | { ( ) { } ( ) } |
| | | Structure | OCOCOC | OOCCOOCC | OOCOCOCC |
| | | Depth | 111111 | 12211221 | 12222221 |
| *Unseen Depth* | Examples with maximum nesting depth strictly greater than two. | Sentence | ( ) { [ { } ] } | [ [ ] ( [ ] ) ] | { } { } ( { { } } ) |
| | | Structure | OCOOOCCC | OOCOOCCC | OCOCOOOCCC |
| | | Depth | 11123321 | 12223321 | 1111123321 |

Table 1: Illustration of different generalization splits. For simplicity, examples are drawn from Dyck-(3, 2). Three sample sentences for each set, their respective sentence structure, and nesting depth are shown above. O and C refer to open and closed brackets in the sentence structure. Note that these are only illustrations of the sampling logic, and the actual experimental results are based on the harder Dyck-(10,20) language, as explained in § 2.1.

# A IMPLEMENTATION DETAILS

## A.1 DYCK DATASET DETAILS

As described in Section 2.1, we sample sentences from Dyck-(10, 20), the language of balanced brackets with 20 bracket types and a maximum nesting depth of 10. We use the sampling distribution described and implemented by Hewitt et al. (2020),[3] following existing work (Yao et al., 2021;

---

[3]https://github.com/john-hewitt/dyckkm-learning

Murty et al., 2023). We insert a special beginning-of-sequence token to the begin of each sequence, and append an end-of-sequence token to the end, and these tokens are included in all calculations of sentence length. Note that we discard sentences with lengths greater than 512. The training set contains 200k sentences and all the generalization sets contain 20k sentences. In all cases, we sample sentences, discarding sentences according to the rules described in Section 2.1, until the dataset has the desired size. See Table 1 for illustration of different generalization sets. Figure 6 plots the distribution of sentence lengths. For reference, we also include an **IID** split, which is created by sampling sentences from the same distribution used to construct the training set but rejecting any strings that appeared in the training set. It turns out that almost all sequences in this subset have unseen structures. (The number of bracket structures at length $n$ is given by the $n^{th}$ Catalan number, so the chance of sampling the same bracket structure twice is very low.) For all of the experiments described in Section 4, results on this subset are nearly identical to results on the *Unseen structure (len > 32)* subset.

## A.2 MODEL DETAILS

For our Dyck experiments, we use a two-layer Transformer, with each layer consisting of one attention head, one MLP, and one layer normalization layer. The model has a hidden dimension of 32, and the attention key and query embeddings have the same dimension. This dimension is chosen based on a preliminary hyperparameter search over dimensions in $\{2, 4, 8, 16, 32, 64\}$ because it was the smallest dimension to consistently achieve greater than 99% accuracy on the IID evaluation split. Each MLP has one hidden layer with a dimension of 128, followed by a ReLU activation. The input token embeddings are tied to the output token embeddings (Press & Wolf, 2017), and we use absolute, learned position embeddings. The model is implemented in JAX (Bradbury et al., 2018) and adapted from the Haiku Transformer (Hennigan et al., 2020). Code for reproducing our results will be made available after the anonymity period.

## A.3 TRAINING DETAILS

We train the models to minimize the cross entropy loss:

$$\mathcal{L} = \frac{1}{|\mathcal{D}|} \sum_{w_{1:n} \in \mathcal{D}} \frac{1}{n} \sum_{i=2}^{n} p(w_i \mid w_{1:i-1}),$$

where $\mathcal{D}$ is the training set, $p(w_i \mid w_{1:i-1})$ is defined according to Section 2.2, and $w_1$ is always a special beginning-of-sequence token. We train the model for 100,000 steps with a batch size of 128 and use the final model for further analysis. We use the AdamW optimizer (Loshchilov & Hutter, 2019) with $\beta_1 = 0.9$, $\beta_2 = 0.999$, $\epsilon = $ 1e-7, and a weight decay factor of 1e-4. We set the learning rate to follow a linear warmup for the first 10,000 steps followed by a square root decay, with a maximum learning rate of 5e-3. We do not use dropout.

# B ADDITIONAL RESULTS

## B.1 CASE STUDY: ANALYZING A DYCK LANGUAGE MODEL

In this section, we attempt to reverse-engineer the algorithm learned by a two-layer Dyck LM by analyzing simplified model representations, using visualization methods that are common in prior work (e.g. Liu et al., 2022; Power et al., 2022; Zhong et al., 2023; Chughtai et al., 2023; Lieberum et al., 2023).

**First layer: Calculating bracket depth** We start by examining the first attention layer. In Fig. 7, we plot an example attention pattern (Fig. 7a), along with the value embeddings plotted on the first two singular vectors (Fig. 7c). As in the human-written algorithm (§2.3), this attention head also appears to (1) attend broadly to all positions, and (2) associate opening and closing brackets with value embeddings with opposite sides. On the other hand, the attention pattern also deviates from the human construction in some respects. First, instead of using uniform attention, the model assigns more attention to the first position, possibly mirroring a construction from Liu et al. (2023). (In preliminary experiments, we found that the model learned a similar attention pattern even when

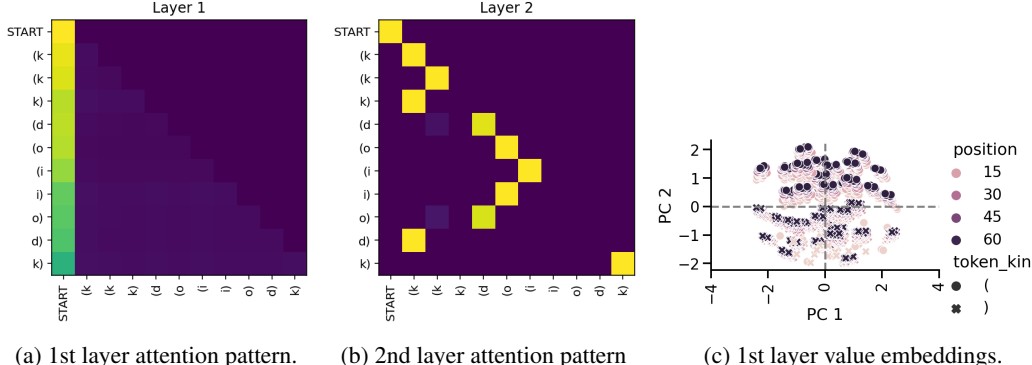

(a) 1st layer attention pattern.  (b) 2nd layer attention pattern  (c) 1st layer value embeddings.

Figure 7: *Left:* An example attention pattern from the first-layer attention head. Each position attends broadly to the full input sequence, but with more attention concentrated on the beginning of sequence token. *Center:* An example attention pattern from the second-layer attention head. Each query assigns the most attention to the most recent unmatched opening bracket. *Right:* The first two singular vectors of the value embeddings. Each point represents the sum of a token embedding and a position embedding (for all token embeddings and positions up to 64).

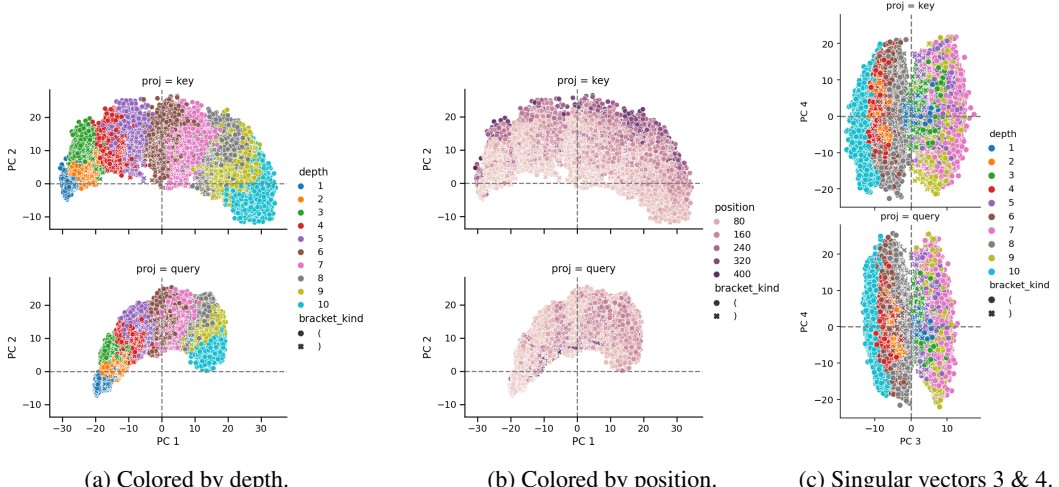

(a) Colored by depth.  (b) Colored by position.  (c) Singular vectors 3 & 4.

Figure 8: Key and query embeddings from the second-layer attention head, projected onto the first four singular vectors and colored by either bracket depth or position.

we did not prepend a START token to the input sequences—that is, the first layer attention head attended uniformly to all positions but placed higher attention on the first token in the sequence.) Second, the value embeddings encode more information than is strictly needed to compute depth. Specifically, Figure 7c shows that the first components of the value embeddings encode position. Coloring this plot by bracket type reveals that the embeddings encode bracket type as well—each cluster of value embeddings corresponds to either opening or closing brackets for a single bracket type. In contrast, in the human-written Transformer algorithm, the value embeddings only encode whether the bracket is an opening or closing bracket and are invariant to position and bracket type This might suggest that the model is using some other algorithm, perhaps in addition to our reference algorithm. Despite these differences, these simplified representations of the model suggest that this model is indeed implementing some form of depth calculation.

**Second layer: Bracket matching** Now we move on to the second attention head. Behaviorally, this attention head appears to implement the bracket-matching function described in §2.3, and which has been observed in prior work (Ebrahimi et al., 2020); an example attention pattern is shown in Fig. 7b. Our goal in this section is to explain how this attention head implements this function in

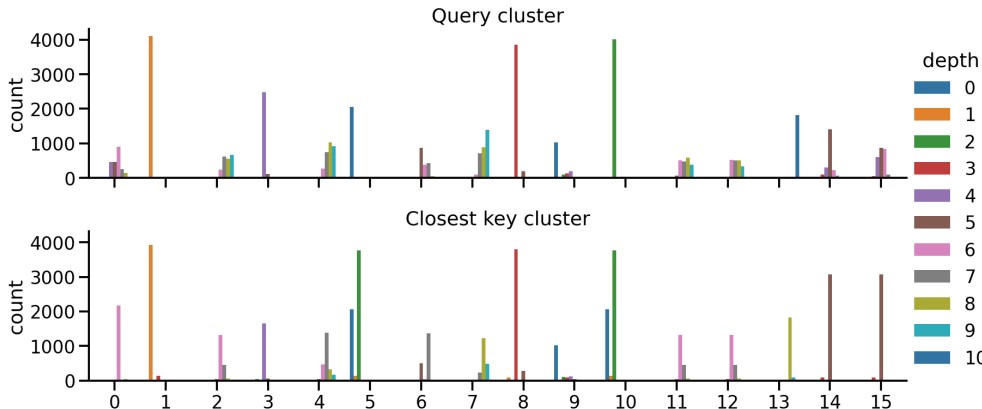

Figure 9: Interpreting the attention mechanism via clustering. The top row plots the depth distribution in each query cluster, and the bottom row plots the depth distribution in the key cluster closest to that query cluster (so some key clusters appear twice). As an interpretation of the model, this figure suggests that the key and query embeddings are reasonably well clustered by depth, and queries generally attend to clusters of the same depth, but the mechanism seems somewhat more effective for some depths (see clusters 1, 3, 8) then others, and in some cases the model might confuse a depth for another depth +/- 2 (clusters 5 and 10). The figure does not capture the mechanism for attending to the most recent token at a matching depth.

terms of the underlying key and query representations. Fig. 8 shows PCA plots of the key and query embeddings from a sample of 1,000 training sequences. Again, these representations resemble the construction from Yao et al. (2021): the direction of the embeddings encodes depth (Fig. 8a), and the magnitude in each direction encodes the position, with later positions having higher magnitudes (Fig. 8b). On the other hand, the next two components (Fig. 8c) illustrate that the model also encodes the parity of the depth, with tokens at odd- and even-numbered depths having opposite signs in the third component. This reflects the fact that matching brackets are always separated by an even number of positions, so each query must attend to a position with the same parity.

Overall, this investigation illustrates how simple representations of the model can suggest an understanding of the algorithm implemented by the underlying model—in this case, that the model implements some version of the depth matching mechanism. (In Appendix B.2, we conduct a similar analysis using clustering and come to similar conclusions; see Fig. 9.)

### B.2  INTERPRETATION BY CLUSTERING

In Figure 9, we plot the distribution of depths associated with each cluster, after applying k-means separately to a sample of key and query embeddings, with k set to 16. Beneath each query cluster, we plot the depth distribution of the nearest key cluster, where distance is defined as the Euclidean distance between cluster centers. This figure illustrates how clustering allows us to interpret the model by means of a discrete case analysis: we can see that the queries are largely clustered by depth, and the nearest key cluster typically consists of keys with the same depth, suggesting that the model implements a (possibly imperfect) version of the depth-matching mechanism described in Section 2.3.

### B.3  ANALYSIS: DEPTH GENERALIZATION

Figure 10 plots key and query embeddings for out-of-distribution data points on the depth generalization split (Section 4.2). The success of the Transformer depends on the model's mechanism for representing nesting depth. To succeed on the depth generalization split, this mechanism must also extrapolate to unseen depths. Figure 4 indicates that the model does extrapolate to some extent, but the simplified models fail to fully capture this behavior. Figure 10 offers some hints about why this might be the case. These plots suggest that there is some systematic generalization (deeper depths are embedded where we might expect). However, we can also visually observe that the deeper depths

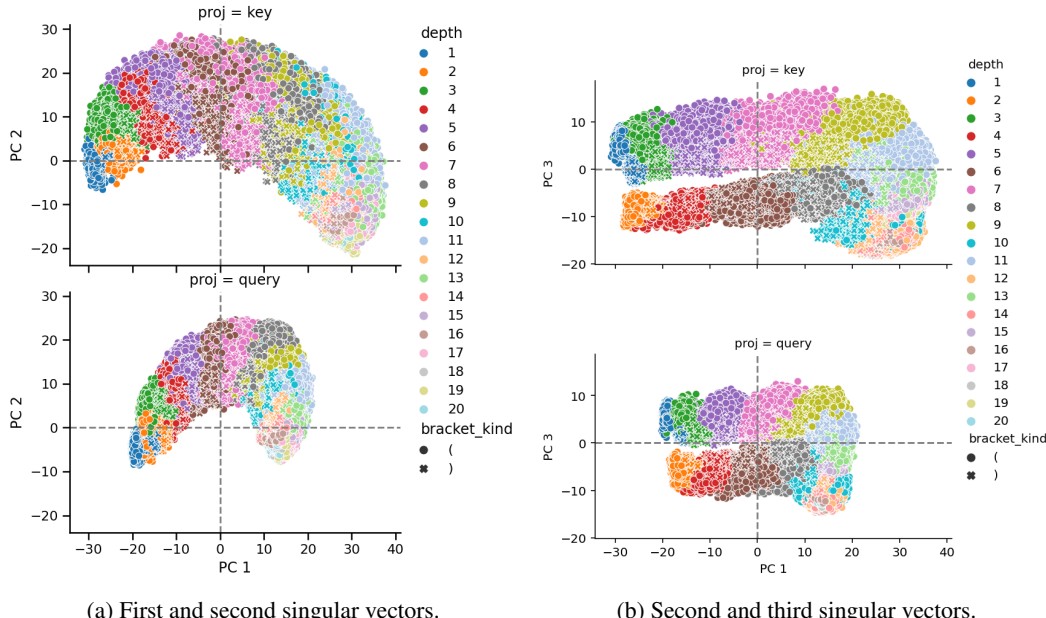

(a) First and second singular vectors.

(b) Second and third singular vectors.

Figure 10: Key and query embeddings for sequences from the out-of-distribution depth generalization test set. Visually, the representations for out-of-distribution depths support some form of depth generalization, but the deeper depths are not separated as effectively, leading to prediction errors.

are less separated, perhaps indicating that the model uses additional directions for encoding deeper depths, but these directions are dropped when we fit SVD on data containing only lower depths. On the other hand, we did not have a precise prediction about where the deeper depths would be embedded, which is a limitation of our black box analysis: without reverse-engineering the lower layers, we cannot make strong predictions beyond the training data.

Are these findings consistent across training runs? In Figure 11, we train a Dyck model with a different random initialization and plot the prediction errors on the depth generalization, recreating Figure 4. The overall pattern is similar in this training run, with the simplified model diverging more from the original model on predictions at deeper query depths.

### B.4    ANALYSIS: STRUCTURAL GENERALIZATION

In Figure 12, we look at the approximation errors on the structural generalization test set (looking at the subset of examples with length $\leq 32$). The figure plots the difference between the position with the highest attention score in the original model and simplified model. In both figures, the simplified version of the model generally attends to an earlier position than the true depth, and that differs from the target depth by an even number of positions. This suggests that these approximations "oversimplify" the model: they capture the coarse grained structure—namely, that positions always attend to positions with the same parity—but understimate the fidelity with which the model encodes depth.

### B.5    ADDITIONAL RESULTS: COMPUTER CODE

**Additional approximation results**   We provide some additional results from the experiments described in Section 5. Figure 13 plots the accuracy of the (original) model at predicting the next character, broken down by the prediction type. The accuracy is relatively high, in part because many characters are whitespace, part of common keywords, or part of variable names that appeared earlier in the sequence. Figure 14 plots the prediction similarity on CodeSearchNet broken down

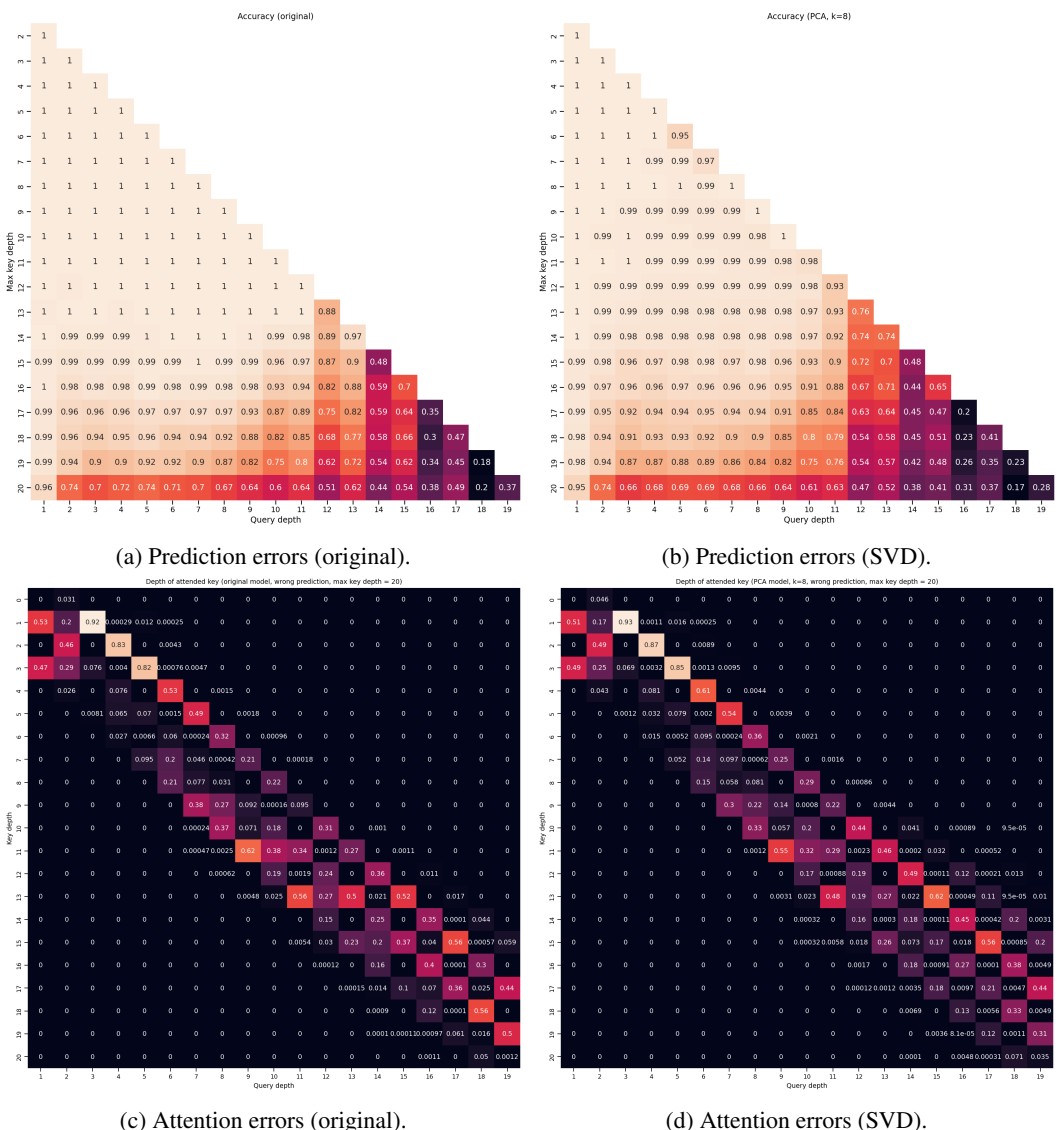

(a) Prediction errors (original).

(b) Prediction errors (SVD).

(c) Attention errors (original).

(d) Attention errors (SVD).

Figure 11: We trained a model on the Dyck dataset with a different random initialization and recreate the plots from Figure 4c. The figure plots the errors of the original model and a rank-8 SVD simplification on the depth generalization test set. Figures 11a and 11b plot the prediction accuracy, broken down by the depth of the query and the maximum depth among the keys. Figures 11c and 11d plot the depth of the token with the highest attention score, broken down by the true target depth, considering only incorrect predictions. The results are similar to the results in Figure 4c, with the simplified model diverging more on predictions with deeper query depths. For example, when the query depth is greater than 15, the lower-dimension model is more likely to attend to keys with a depth four less than the target depth.

by whether the underlying model's prediction is correct or incorrect. Prediction similarity is lower overall when the model is incorrect, although the gap between in-domain and out-of-domain approximation scores is generally smaller. Figure 15 breaks down the results by prediction type, showing the effect of simplifying each attention head, aggregated across models trained with three random initializations, each with four layers and four attention heads per layer. The findings observed in Figure 5b are generally consistent across training results, with the the gap between in-distribution and out-of-distribution approximation scores varying depending on the prediction type, although using lower dimensions leads to more consistent generalization gaps across categories.

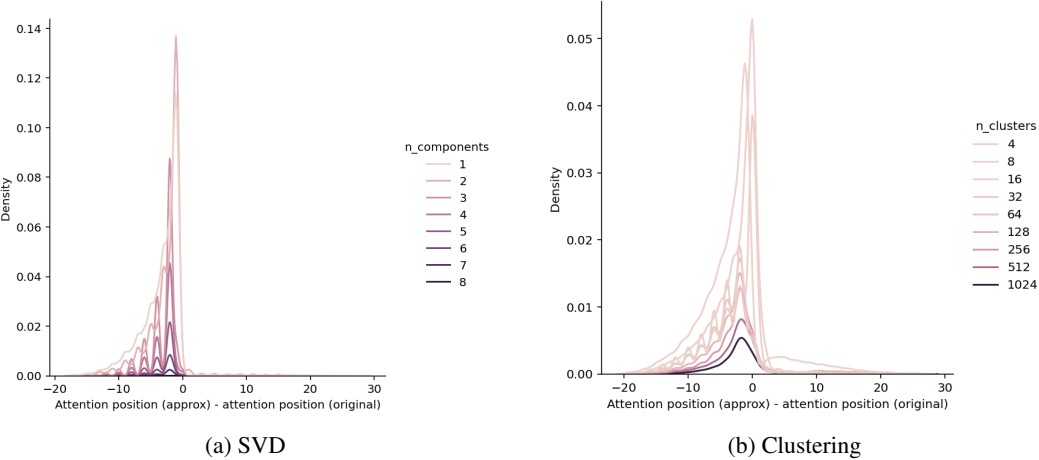

(a) SVD        (b) Clustering

Figure 12: These figures illustrate the kinds of errors in attention patterns that the approximations introduce on Unseen structure (length $\leq 32$) evaluation split. They plot the distribution of distances between the position to which the original model assigns the most attention, and the position attended to by the simplified model. Both plots have a periodic structure, where the simplified model attends to a position that differs from the correct position by a multiple of two. This error pattern can be understood by noting that the parity of the nesting depth (which is equivalent to the parity of the position) is a prominent feature in the key and query embeddings (Figure 8c).

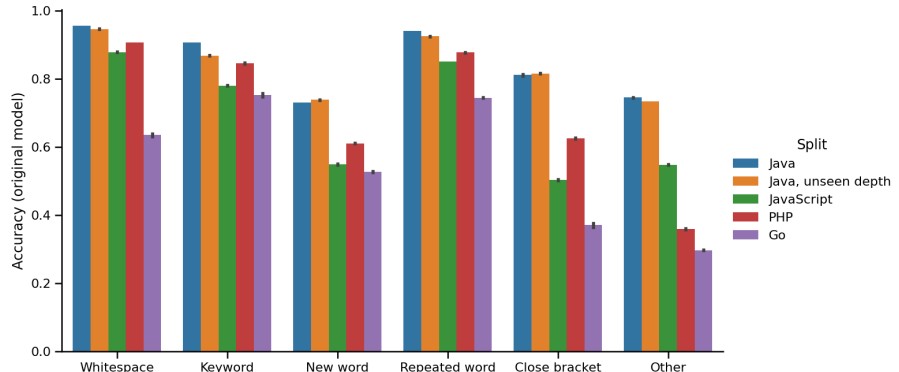

Figure 13: Accuracy at predicting the next character in CodeSearchNet, broken down by prediction type and evaluated on different generalization splits, averaged over three training runs. See Section B.5 for more details.

**Which attention head is associated with the biggest generalization gap?** In Figure 16, we plot the generalization gap for each attention head in a code completion model, comparing the difference in the Same Prediction approximation score between Java examples and non-Java examples, using a rank-16 SVD approximation for the given key and query embeddings. For most categories, the attention head associated with the largest gap is the fourth head in the final layer. In Figure 17, we plot two example attention patterns from this head. From the attention pattern, this attention head appears to implement part of an "induction head" circuit (Elhage et al., 2021).

## B.6 ADDITIONAL DATASET: SCAN

To assess whether our findings generalize to other settings, we also train models on the SCAN dataset (Lake & Baroni, 2018). SCAN is a synthetic, sequence-to-sequence semantic parsing dataset designed to test systematic generalization. The input to the model is an instruction in semi-natural language, such as "turn thrice", and the target is to generate a sequence of executable commands ("TURN TURN TURN").

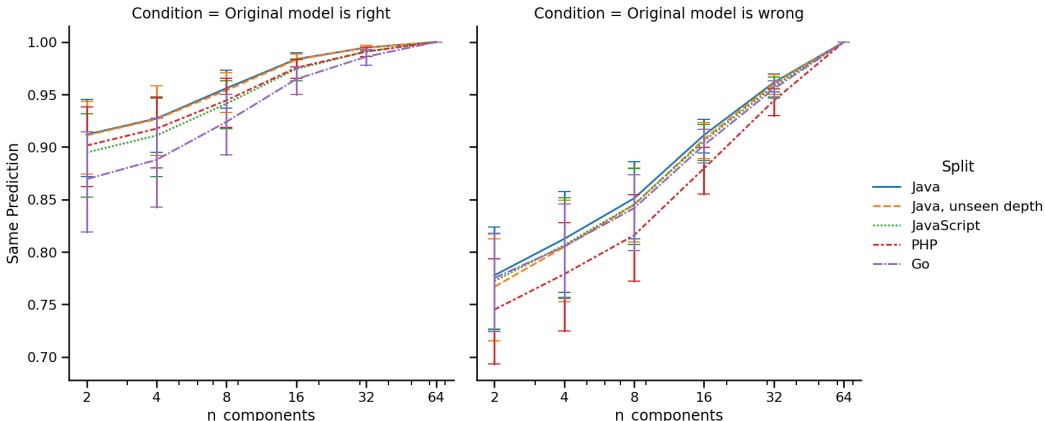

Figure 14: Predicting closing brackets in different versions of CodeSearchNet after reducing the dimension of the key and query embeddings using SVD. We apply dimension reduction to each attention head independently and aggregate the results over attention heads and over models trained with three random seeds. The results are partitioned according to whether the original model makes the correct (*top*) or incorrect (*bottom*) prediction. The prediction similarity between the original and simplified models is consistently higher on in-distribution examples (*Java, seen depth*) relative to examples with deeper nesting depths or unseen languages. The gap is somewhat larger when we resample the bracket types and increase the number of bracket types.

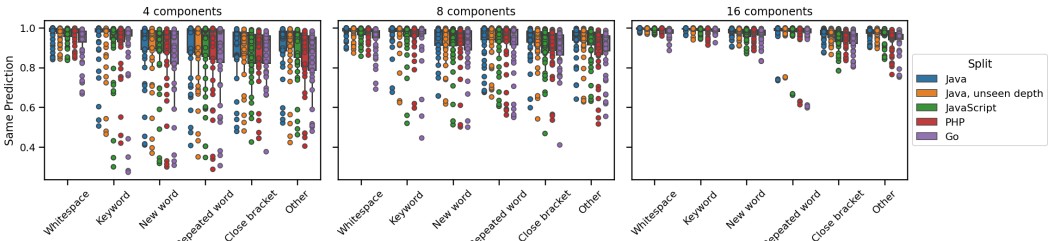

Figure 15: Prediction similarity on CodeSearchNet after reducing the dimension of the key and query embeddings using SVD, filtered to the subset of tokens that the original model predicts correctly. We train models with three random initializations. Each model has four layers, four attention heads per layer, and an attention embedding size of 64, and we apply dimension reduction to each attention head independently. This figure depicts the results from all three models using different numbers of SVD components, broken down by prediction type. Each point on the plot shows the prediction similarity after simplifying one attention head. The findings in 5b are generally consistent across training results, with the the gap between in-distribution and out-of-distribution approximation scores varying depending on the prediction type, although using lower dimensions leads to more consistent generalization gaps across categories.

**Data** We consider two generalization settings. First, we test models on the **Add jump** generalization split, in which the test examples are compositions using a verb (jump) that appears in the training set only as an isolated word. We train this model on the training set introduced by Patel et al. (2022),[4] which augments the original training data with more verbs, as this was shown to enable Transformers to generalize to the *Add jump* set. Second, we test models on **Length** generalization. We use the length generalization splits from Newman et al. (2020),[5]. The examples in the training set are no longer than 26 tokens long, and generalization splits have examples with lengths spanning from 26 tokens to 40 tokens.

---

[4]https://github.com/arkilpatel/Compositional-Generalization-Seq2Seq
[5]https://github.com/bnewm0609/eos-decision

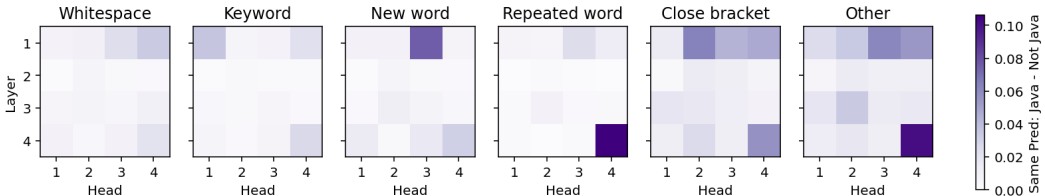

Figure 16: The generalization gap for each attention head in a model for CodeSearchNet, defined as the difference between the Same Prediction score measured on in-domain examples (Java) compared to samples in unseen languages, using a rank-16 SVD approximation for the key and query embeddings. Simplifying the fourth head in the fourth layer leads to a generalization gap for most prediction types, with the biggest gap on predicting words that appeared earlier in the sequence. This head seems to form part of an "induction head" mechanism (see Fig. 17).

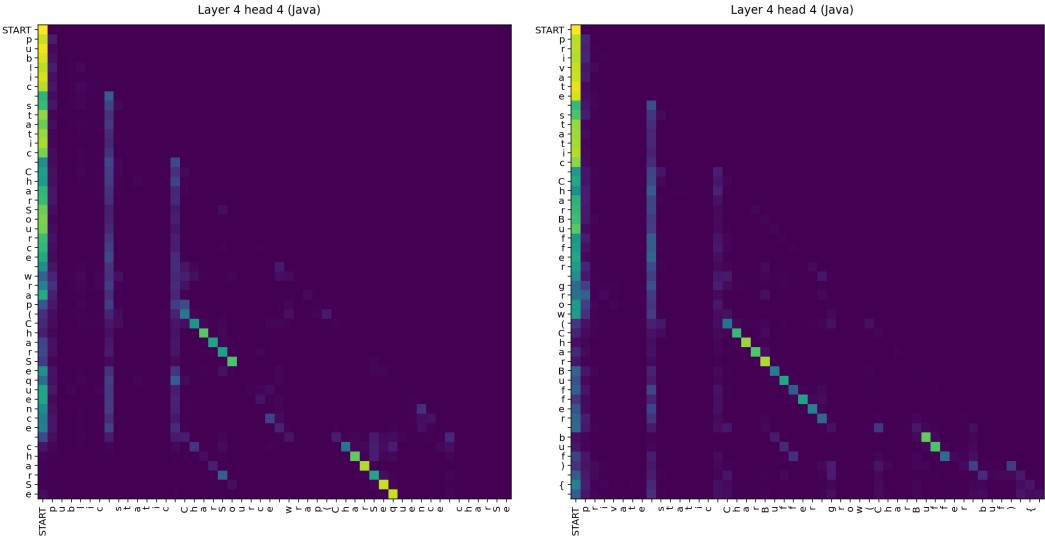

Figure 17: Two example attention patterns from the attention head associated with the largest generalization gap on a code completion task (see Fig. 16). Each row represents a query and each column represents a key, and we show the first 50 characters of two Java examples. This attention head appears to implement the "induction head" pattern, which increases the likelihood of generating a word that appeared earlier in the sequence.

**Model and training** As in our Dyck experiments, we use a decoder-only Transformer. In the *Add jump* setting, we modify the attention mask to allow the model to use bi-directional attention for the input sequence, and we train the model to generate the target tokens. In the *Length* generalization setting, we use an autoregressive attention mask at all positions, and no position embeddings, because this has been shown to improve the abilities of Transformers to generalize to unseen lengths (Kazemnejad et al., 2023). We conduct a hyper-parameter search over number of layers (in $\{2, 3, 4, 6\}$) and number of attention heads (in $\{1, 2, 4\}$) and select the model with the highest accuracy on the generalization split, as our main goal is to evaluate simplified model representations in settings where the underlying model exhibits some degree of systematic generalization. The hidden dimension is 32, and the dimension of the attention embeddings is 32 divided by the muber of attention heads. For *Add jump*, this is a three layer model with two attention heads, which achieves an OOD accuracy of around 99%. For *Length*, this is a six layer model with four attention heads, achieving an OOD accuracy of around 60%. We use a batch size of 128, a maximum learning rate of 5e-4, and a dropout rate of 0.1, and train for 100,000 steps. All other model and training details are the same as in Appendix A.2 and A.3.

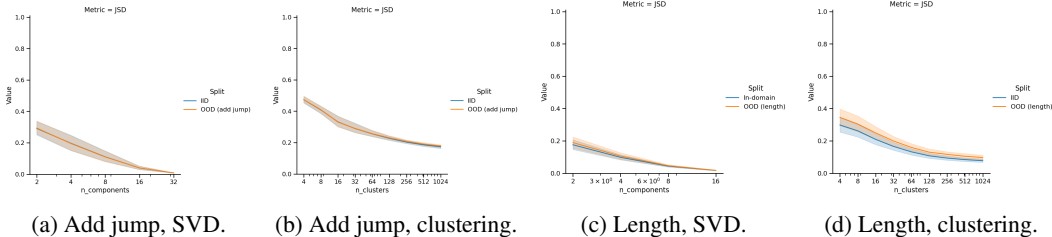

(a) Add jump, SVD.  (b) Add jump, clustering.  (c) Length, SVD.  (d) Length, clustering.

Figure 18: Attention approximation metrics for models trained on the SCAN dataset, evaluated on two systematic generalization splits: the *Add jump* evaluation set, and the *Length* generalization set, which contains examples with unseen lengths. We apply these approximations to each attention head individually, and plot the results and 95% confidence interval averaged over the heads. See Appendix B.6.

**Results** After training the models, we evaluate the effect of dimension-reduction and clustering, applied to the key and query embeddings. We apply this simplification independently to each head in the model and report the average results in Figure 18. On the *Add jump* generalization set, there is no discernible generalization gap. This result is in line with a finding of Patel et al. (2022), who found that, in models that generalize well, the embedding for the *jump* token is clustered with the embeddings for the other verbs, which would suggest that simplifications calculated from training data would also characterize the model behavior on the *Add jump* data. On the *Length* generalization set, there is a small but consistent generalization gap for both dimension reduction and clustering. These results suggest that generalization gaps can also appear in other settings, and underscore the value of evaluating simplifications on a variety of distribution shifts. On the other hand, we do not attempt to conduct a mechanistic interpretation of these larger models; further investigations are needed to understand the implications of these (small) generalization gaps for model interpretation.

