# OpenReview forum: "Interpretability Illusions in the Generalization of Simplified Models"
_ICLR.cc/2024/Conference — Submitted to ICLR 2024_

### Official Review · Reviewer_5xMH · 2023-10-27

**Soundness:** 3 good
**Presentation:** 3 good
**Contribution:** 1 poor
**Rating:** 3
**Confidence:** 3

**Summary:**

The paper offers an analysis of the extrapolation performance of several simplified representation methods used to understand the information processing of deep architectures. The analysis is solely focused on Transformers trained on Dyck-k languages, which consists of strings of matched brackets of k different types. These models are tested in and out of distribution by manipulating the maximum hierarchical depth and other parameters. The study considers two simplified models: PCA and k-means clustering of the key and query embeddings of the transformer layers. The authors conclude that these simple methods offer a good description of the model in-sample, but fail to explain the behavior of the model out-of-sample.

**Strengths:**

- The paper is well written and it contains several interesting considerations on the nature and interpretation of transformers.
- The problem area is vitally important to the implementation of AI in real-world applications, and the focus on out-of-sample performance is interesting and well motivated.
- The experimental analysis is detailed and rigorous.

**Weaknesses:**

The focus of the experimental analysis is too narrow. The introduction section does a good job in outlining the research goals, but this aim is then overly specialized to a specific class of models trained on a toy problem. It is therefore very difficult to extrapolate the conclusions of the paper outside of its narrow domain, which in itself is not very useful for the broader literature.

All in all, in spite of the several interesting insights present in the text, the contribution and novelty of this work is very limited.

**Questions:**

- Can you include the analysis of other datasets and architectures? I do appreciate your work on toy-languages. However, I would like to see these insights to be applied  to models trained on naturalistic data.

---

> ### Author Response · Authors · 2023-11-22
> **Author response to reviewer 5xMH**
>
> Thank you for your helpful feedback. We are glad that you find our paper addresses an important problem, the experiments are detailed and rigorous, the writing is clear, and the paper presents interesting insights.
>
> > The focus of the experimental analysis is too narrow
>
> As described in the overall response, we have extended our experiments to include a more realistic setting of language modeling in code datasets. We originally chose to focus on the Dyck languages because they capture a fundamental aspect of natural languages–hierarchical syntactic structure–but are simple enough to allow for some mechanistic understanding. The Dyck languages are considered a canonical instance of context free languages, because any context free language can be formed by the intersection of Dyck and a regular language [1]. For this reason, a variety of prior work has also focused exclusively on this setting as a controlled test bed for studying questions about expressivity and interpretability (e.g. [2-4]).
>
> However, as noted in the general response, we have extended our experiments to incorporate an additional setting: language modeling on code datasets, training models with more layers and attention heads. These experiments show that generalization gaps also occur in more naturalistic settings and provide some additional insight into the relationship between generalization gaps and properties of the task. To summarize:
>
> * We trained character-level language models on Java functions and evaluated generalization to functions with deeper nesting depths and unseen programming languages.
> * These experiments revealed a consistent generalization gap: simplified proxy models are more faithful to the original model on in-distribution data and worse on OOD data.
> * We also broke down these results according to different subsets of the code completion task, and found that the generalization gap is notably larger on some types of predictions. In particular, the generalization gap is larger on subtasks that can be seen as more “algorithmic”, including predicting closing brackets and copying variable names from earlier in the sequence. The gap is smaller when predicting whitespace characters and keywords, perhaps because these predictions rely more on local context than precise attention.
> * We were able to attribute some of these generalization gaps to an “induction head” mechanism, which copies words from earlier in the sequence. Simplifying this head leads to large gaps in approximation quality between in-domain and out-of-domain samples, meaning that the low-dimensional approximation underestimates the extent to which the induction head mechanism will generalize to unseen distributions.
>
> Overall, these experiments provide evidence that generalization gaps also occur in naturalistic settings, and they provide additional insight into how these gaps are related to different properties of sequence modeling tasks. Specifically, generalization gaps might be most pronounced in “algorithmic” settings where the model must use a particular feature in a precise, context-dependent way, and there are fewer or no other proxies of that feature available in the data. The effect is diminished in settings where a variety of local features contribute to the model’s prediction. We hope that these additional experiments help to address concerns about the scope of our analysis, and, if so, you might consider reevaluating your score.
>
> > Specialized to a specific class of models
>
> We note that our additional experiments extend our analysis to models with more layers and attention heads. We do focus on a specific class of models–Transformer language models. However, this is now the predominant class of models across AI, and we are joining a considerable body of research in investigating how we can better understand these systems.
>
> > References
>
> [1] Kambites, M. (2009). Formal languages and groups as memory. Communications in Algebra, 37(1), 193-208.
>
> [2] Yao, S., Peng, B., Papadimitriou, C., & Narasimhan, K. (2021). Self-attention networks can process bounded hierarchical languages. arXiv preprint arXiv:2105.11115.
>
> [3] Hewitt, J., Hahn, M., Ganguli, S., Liang, P., & Manning, C. D. (2020). RNNs can generate bounded hierarchical languages with optimal memory. arXiv preprint arXiv:2010.07515.
>
> [4] Wen, K., Li, Y., Liu, B., & Risteski, A. (2023). (Un) interpretability of Transformers: a case study with Dyck grammars.

---

### Official Review · Reviewer_H66k · 2023-10-30

**Soundness:** 3 good
**Presentation:** 3 good
**Contribution:** 3 good
**Rating:** 6
**Confidence:** 4

**Summary:**

The paper describes how much the simplified transformer models represent the behaviour of the original ones.

The authors consider a use case of Dyck balanced-parenthesis languages and show that while the simplified proxies, using hard (one-hot) attention, show alignment with the behaviour of the original models, they do not match the behaviour out-of-distribution. They use the evaluation methodology as per Murthy et al (2023) which involves predicting closing brackets at least ten positions away from the corresponding opening brackets and evaluating the  highest-likelihood prediction accuracy.

**Strengths:**

Pros:
- (Originality) The originality of the paper stems from analysing the claims of interpretability

- (Significance) It is important to see the detailed analysis of limitations of simplified models on a simple example which would highlight the deficiencies of such models.

- (Quality) The paper thoroughly addresses reproducibility

- (Clarity) The paper is clearly written (however, see Q1-Q3)

**Weaknesses:**

Cons:

- (Elements of significance) Given that the analysis focuses on Dyck balanced-parenthesis languages, it is largely limited and arguably backs up the intuitive claim that the simplified models do not fully represent the behaviour of the original models; however, I still think that it  is still significant because we have such evidence described in detail as it helps inform large-scale model interpretability studies

**Questions:**

1. In Figure 6, it seems like for the longer key depths, the predictions diverge more. Would the authors be able to emphasise more whether it is always the case and if there are any solid reasons behind this particular behaviour?

2. In Figure 1 description it is stated that ‘On the depth generalization split, the models achieve approximately 80% accuracy.’ Is it 80% or around 75% as can be seen in the purple curve on the image? (It does not affect any conclusion, just found that I could not fully explain this discrepancy)

3. “However, the error patterns diverge on depths greater than ten, suggesting that the lower-dimension model can explain why the original model makes mistakes in some in-domain cases, but not out-of-domain” To what extent does it  happen consistently across different training trajectories of the stochastic gradient descent and/or across different datasets? In other words, plausible it sounds, would the same tendency repeat if we change the data or train the model again?

---

> ### Author Response · Authors · 2023-11-22
> **Author response to reviewer H66k**
>
> Thank you for your helpful feedback! We’re glad you see the paper as addressing an important problem, found the experiments to be original and thoroughly reproducible, and found the paper to be clearly written.
>
> > Limited scope
>
> As noted in the general response, we have extended our experiments to incorporate an additional setting: language modeling on code datasets, training models with more layers and attention heads. These experiments show that generalization gaps also occur in more naturalistic settings and provide some additional insight into the relationship between generalization gaps and properties of the task. To summarize:
>
> * We trained character-level language models on Java functions and evaluated generalization to functions with deeper nesting depths and unseen programming languages.
> * These experiments revealed a consistent generalization gap: simplified proxy models are more faithful to the original model on in-distribution data and worse on OOD data.
> * We also broke down these results according to different subsets of the code completion task, and found that the generalization gap is notably larger on some types of predictions. In particular, the generalization gap is larger on subtasks that can be seen as more “algorithmic”, including predicting closing brackets and copying variable names from earlier in the sequence. The gap is smaller when predicting whitespace characters and keywords, perhaps because these predictions rely more on local context than precise attention.
> * We were able to attribute some of these generalization gaps to an “induction head” mechanism, which copies words from earlier in the sequence. Simplifying this head leads to large gaps in approximation quality between in-domain and out-of-domain samples, meaning that the low-dimensional approximation underestimates the extent to which the induction head mechanism will generalize to unseen distributions.
>
> Overall, these experiments provide evidence that generalization gaps also occur in naturalistic settings, and they provide additional insight into how these gaps are related to different properties of sequence modeling tasks. Specifically, generalization gaps might be most pronounced in “algorithmic” settings where the model must use a particular feature in a precise, context-dependent way, and there are fewer or no other proxies of that feature available in the data. The effect is diminished in settings where a variety of local features contribute to the model’s prediction.
>
> > Why do predictions diverge more on longer key depths?
>
> While we don’t have a precise mechanistic explanation for this result, we think our observations suggest some hypotheses (in particular, see Appendix Figure 10 in our revision). Specifically, the success of the Transformer depends on the mechanism for representing nesting depth. To succeed on the depth generalization split, this mechanism must also extrapolate to unseen depths. Figure 10 suggests that the original model does extrapolate to some extent, but the simplified models fail to fully capture the mechanism for extrapolating to deeper depths. This could lead to two kinds of error. First, even if the query token appears at a seen depth, there might be deeper key tokens earlier in the sequence, and if these keys are not represented well, they might act as “distractors” for shallower queries. Second, if the query token appears at an unseen depth, the model might fail to match it to the correct bracket if the representations for these depths are not well-separated. In the error patterns in Figure 4 (Figure 6 in our original draft) we see evidence that both types of errors occur, and the divergences are greater when the query token is at an unseen depth.
>
> > Are these results consistent across training trajectories?
>
> In the appendix, we have added a figure showing the error patterns for another model trained with a different random initialization (Figure 11). The general trends are similar, with predictions diverging more on queries appearing at unseen depths. We would also note that, in our additional experiments on code completion, we reported the results from models trained with multiple random initializations and found that different training runs lead to similar trends in terms of generalization gaps.
>
> > Description of Figure 1 (accuracy on depth generalization)
>
> Thank you for pointing this out! This was an oversight and we have corrected the description of the figure in our revision.

---

> ### Comment · Reviewer_H66k · 2023-11-23
>
> Many thanks! I had a look at the experiments and responses, and it appears to improve the paper and address the reviewers' concerns. I am saying 'appears to improve' because I would like to read it more thoroughly,  and there has been unfortunately only a short period of time between the rebuttal submission and the deadline. I understand this is due to the very limited time constraints for the rebuttal preparation, and that it took lots of effort to prepare the revision and I would like to thank the authors for it.
>
>  Therefore, I will read it carefully again and take these comments and revision into account when submitting post-rebuttal recommendation. My recommendation remains unchanged for now.

---

### Official Review · Reviewer_nnFq · 2023-10-31

**Soundness:** 3 good
**Presentation:** 2 fair
**Contribution:** 3 good
**Rating:** 6
**Confidence:** 2

**Summary:**

The paper presents an analysis of transformer models trained on the Dyck languages. To do so, the models are simplified and analyzed with data-dependent and data-independent tools, highlighting a discrepancy between the behaviour of original models and simplified models on out-of-distribution data.

**Strengths:**

The paper provides several methods to analyse transformers trained on the Dyck language, investigating whether simplified versions of the model are faithful to the original one on out-of-distribution test sets.

**Weaknesses:**

Being unfamiliar with the literature, it is hard for me to understand the point of the analysis, and it is hard to tell whether that is due to a poor presentation or due to my lack of understanding. However, what I find a weakness of the paper is the fact that the analysis is not paired with proposed improvements or solutions. For example, what do the results from the paper entail? Is it that transformer models are not suitable for learning language models? Or is it that using model simplifications, while facilitating the analysis of some properties of the model, leads to a mismatch with the original model on out-of-distribution samples? If so, what would be a better way to analyze transformer models, to avoid the found shortcomings of current methods?

**Questions:**

In addition to the questions in Weaknesses:

- What conclusions can be drawn from Deep Learning practitioners? Are transformers reliable for learning languages?
- Does the analysis extend also to similar models or other datasets?

---

> ### Comment · Reviewer_5xMH · 2023-11-13
> **Concerns about quality of the review**
>
> I find quite concerning that the reviewer is proposing a rejection based on its own lack of understanding. If you do not feel qualified to review a paper, you should communicate it to the AC and ask to  be dispensed.

---

> ### Author Response · Authors · 2023-11-22
> **Author response to reviewer nnFq**
>
> Thank you for your feedback.
>
> > The analysis is not paired with proposed improvements or solutions
>
> We would like to emphasize that our aim in this paper isn’t to present new methods for analyzing Transformer language models. Our goal is to better understand the limitations of interpretability methods that are commonly used, which try to understand models by way of simplified representations. It is important to note that practitioners may use this approach without recognizing the implicit assumption that “the simplified and original model respond similarly to distribution shifts.” Our main point is to illustrate how simplified proxy models may be faithful to the original model on in-distribution data but fail to explain the model’s behavior on out-of-distribution data, giving rise to interpretability illusions. We conduct further analysis to characterize how the faithfulness of these explanations is influenced by the choice of simplification, the underlying task, the required features for solving the problem, the form of distribution shift, and the model's performance. Our findings also provide a counterpoint to a common assumption about the relationship between simplification and generalization in machine learning, highlighting settings where simplified versions of a model incur a larger performance drop on tests of systematic generalization.
>
> We agree with all reviewers (iDw5, oEsb, H66k, 5xMH) that this is an important question. We argue that extensive analyses are needed to disentangle the role of these variables, investigate and explain the underlying reasons, and effectively convey the message. We believe this is an important contribution, and the best way to share these results with the community is through an “insight and analysis paper”. While a “method paper” could be a valuable future step, it serves a different purpose and may not be the most suitable platform for sharing this particular insight.
>
> > What conclusions can be drawn from Deep Learning practitioners? Are transformers reliable for learning languages?
>
> The main takeaways for practitioners are to be aware of limitations to an approach to model interpretation: simplified proxy models might be faithful to a model on some distributions but fail to predict how the model will generalize to different distributions. Our analysis is not focused on the question of whether or not Transformers are reliable for learning languages (although we think our findings might provide some information about the extent to which we can expect Transformers to generalize to different kinds of distribution shifts, such as to deeper hierarchical structures).
>
> > Does the analysis extend also to similar models or other datasets?
>
> As noted in the general response, we have extended our experiments to incorporate an additional setting: language modeling on code datasets, training models with more layers and attention heads. These experiments show that generalization gaps also occur in more naturalistic settings and provide some additional insight into the relationship between generalization gaps and properties of the task. To summarize:
>
> * We trained character-level language models on Java functions and evaluated generalization to functions with deeper nesting depths and unseen programming languages.
> * These experiments revealed a consistent generalization gap: simplified proxy models are more faithful to the original model on in-distribution data and worse on OOD data.
> * We also broke down these results according to different subsets of the code completion task, and found that the generalization gap is notably larger on some types of predictions. In particular, the generalization gap is larger on subtasks that can be seen as more “algorithmic”, including predicting closing brackets and copying variable names from earlier in the sequence. The gap is smaller when predicting whitespace characters and keywords, perhaps because these predictions rely more on local context than precise attention.
> * We were able to attribute some of these generalization gaps to an “induction head” mechanism, which copies words from earlier in the sequence. Simplifying this head leads to large gaps in approximation quality between in-domain and out-of-domain samples, meaning that the low-dimensional approximation underestimates the extent to which the induction head mechanism will generalize to unseen distributions.
>
> Overall, these experiments provide evidence that generalization gaps also occur in naturalistic settings, and they provide additional insight into how these gaps are related to different properties of sequence modeling tasks. Specifically, generalization gaps might be most pronounced in “algorithmic” settings where the model must use a particular feature in a precise, context-dependent way, and there are fewer or no other proxies of that feature available in the data. The effect is diminished in settings where a variety of local features contribute to the model’s prediction.

---

> > ### Comment · Reviewer_nnFq · 2023-11-23
> >
> > I thank the reviewers for answering my questions and for improving the clarity of the manuscript, as well as extending the experimental section. This helped me get a better understanding of the scope and contribution of this work. I have updated my score.

---

### Official Review · Reviewer_oEsb · 2023-11-01

**Soundness:** 4 excellent
**Presentation:** 4 excellent
**Contribution:** 3 good
**Rating:** 8
**Confidence:** 4

**Summary:**

The paper provides a case study of using simplified model to interpret a trained transformer model on an algorithmic task. It's shown that using dimension reduction or clustering to simplify the model down to a proxy model may yield interpretability illusion. In particular, the simplified proxy model is not faithful in out-of-distribution settings, and cannot be used to reliably predict the original model's OOD error.

**Strengths:**

The paper is focused on a classic formal language task (Dyck grammer) and provides a convincing case study of interpretability illusion in Transformer language model. I think the main observation from the paper is interesting and quite relevant.

The distributions are novel, as prior work in mechanistic interpretability mostly gives positive results.

I think the main result is surprising, where the simplified model generalizes less well to OOD data than the full model, where intuitions from learning theory would suggest the opposite.

The paper is well-written and the illustrations are clear.  It also gives a good survey of related work.

**Weaknesses:**

While the paper delivers a strong conceptual message, at a technical level, it is a single case study on a somewhat toy algorithmic task. That is, the scope of the work is a bit limited. I personally would be interested in a broader study on similar formal language tasks (for example, on other languages expressed by finite-state automata https://arxiv.org/abs/2210.10749).

The paper would also be stronger if it looks into why the simplified model generalizes less well to the depth split. Figure 6 is an interesting observation. What is really being truncated by SVD (which plays a role for OOD generalization)? Is there any mechanistic story here?

Minor suggestions
---

Figure 2(a)(b) should be accompanied by a color scale. Does yellow indicate 1 and green indicate something less than 1?

Also, for Figure 2(a) and related experiments, if you don’t prepend the START symbol, what would the attention pattern look like? Does that affect any of your results here?

“Second, the value embeddings encode more information than is strictly needed to compute depth, which might suggest that the model is using some other algorithm” — Can you expand on this? What is the extra information, if you have looked into it all?

**Questions:**

I have asked a few questions above.

---

> ### Author Response · Authors · 2023-11-22
> **Author response to Reviewer oEsb**
>
> Thank you for your helpful feedback! We are glad that you find our results to be interesting, surprising, and relevant, and you find our paper well-written and clear.
>
> > Limited scope
>
> As noted in the general response, we have extended our experiments to incorporate an additional setting — language modeling on code datasets. These experiments show that generalization gaps also occur in more naturalistic settings and provide some additional insight into the relationship between generalization gaps and properties of the task. To summarize:
>
> * We trained character-level language models on Java functions and evaluated generalization to functions with deeper nesting depths and unseen programming languages.
> * These experiments revealed a consistent generalization gap: simplified proxy models are more faithful to the original model on in-distribution data and worse on OOD data.
> * We also broke down these results according to different subsets of the code completion task, and found that the generalization gap is notably larger on some types of predictions. In particular, the generalization gap is larger on subtasks that can be seen as more “algorithmic”, including predicting closing brackets and copying variable names from earlier in the sequence. The gap is smaller when predicting whitespace characters and keywords, perhaps because these predictions rely more on local context than precise attention.
> * We were able to attribute some of these generalization gaps to an “induction head” mechanism, which copies words from earlier in the sequence. Simplifying this head leads to large gaps in approximation quality between in-domain and out-of-domain samples, meaning that the low-dimensional approximation underestimates the extent to which the induction head mechanism will generalize to unseen distributions.
>
> Overall, these experiments provide evidence that generalization gaps also occur in naturalistic settings, and they provide additional insight into how these gaps are related to different properties of sequence modeling tasks. Specifically, generalization gaps might be most pronounced in “algorithmic” settings where the model must use a particular feature in a precise, context-dependent way, and there are fewer or no other proxies of that feature available in the data. The effect is diminished in settings where a variety of local features contribute to the model’s prediction.
>
> > Why does the simplified model generalize less well to the depth generalization split?
>
> We don’t have a precise mechanistic explanation for this result but we think our observations suggest some hypotheses (in particular, see Appendix Figure 10 in our revision). Specifically, the success of the Transformer depends on the mechanism for representing nesting depth. Each depth has to be represented as a separate direction, to be used in the second attention layer. To succeed on the structural generalization split, the simplified model just needs to capture the in-distribution behavior of this mechanism–embedding seen depths. But to succeed on the depth generalization split, this mechanism must also extrapolate to unseen depths. Figure 10 suggests that the model does extrapolate to some extent, but the simplified models fail to fully capture this behavior. For example, perhaps the model uses additional directions for encoding deeper depths, but these directions are dropped when we fit SVD on data containing only lower depths. We have updated Appendix B.3 to include this discussion.
>
> > What if we don’t prepend a START token?
>
> In preliminary experiments, we found that the model learned a similar attention pattern even when we didn’t prepend a START token – that is, the first layer attention head attended uniformly to all positions but placed higher attention on the first token in the sequence. This pattern resembles a construction from [1]: the first token is used to absorb extra attention so that the normalizing constant in the attention output is consistent across sequence lengths. We have noted this in Appendix B.1 in our revision.
>
> [1] Liu et al., 2022. Transformers Learn Shortcuts to Automata.
>
> > What extra information is encoded in the value embeddings?
>
> From plotting the value embeddings (Figure 7c in the revision), we can see that the first components of the value embeddings encode position. The value embeddings also encode bracket type – each cluster of value embeddings corresponds to either opening or closing brackets for a single bracket type. In contrast, in the human-written Transformer algorithm, the value embeddings only encode whether the bracket is an opening or closing bracket and are invariant to position and bracket type. We have updated section B.1 to explain this point.

---

> > ### Comment · Reviewer_oEsb · 2023-11-22
> >
> > Thank you for responding to my original review! I read the revision and have updated my rating to 8.

---

### Official Review · Reviewer_iDw5 · 2023-11-01

**Soundness:** 2 fair
**Presentation:** 2 fair
**Contribution:** 2 fair
**Rating:** 5
**Confidence:** 2

**Summary:**

This study focuses on the question whether a simplified model (e.g., models obtained from dimensionality reduction or clustering) can still faithfully mimic the behavior of the original model on out-of-distribution data. This study conducts experiments on synthetic datasets constructed using the Dyck balanced-parenthesis language and shows that simplified models are less faithful on out-of-distribution data compared to in-distribution data and can under- or overestimate the generalization ability of the original model.

**Strengths:**

The paper focuses on a very important question, that is, whether the explanation model/proxy is faithful to the original model. Or more precisely, whether the explanation model can mimic the original model’s behavior on different data distributions. Some existing explanation methods, such as distilling the target model into a decision tree, cannot guarantee the faithfulness on out-of-distribution data (e.g., masked input samples). Therefore, it is of significant value to delve into this issue.

**Weaknesses:**

1.	I’m not familiar with the Dyck balanced-parenthesis language used in this paper, so I feel a bit confused and overwhelmed reading Section 2.1. It would be a great help if the authors can give some toy examples when introducing the Dyck languages.
2.	The phrase “simplified models” can be misleading in this paper’s context. I was thinking of methods such as knowledge distillation or network pruning when I first see the phrase “simplified models”. However, what the paper mainly focuses on are dimensionality reduction and clustering methods. It is encouraged to use a more precise word other than “simplified”.

**Questions:**

I have several confusions when reading the paper, and hope these can be resolved by the authors’ rebuttal.

1.	In Section 2.1, the authors construct different testing datasets (named as **IID, Seen struct, Unseen struct (len <= 32), Unseen struct (len > 32), and Unseen depth**, respectively) to evaluate the model’s generalization ability. I wonder which of these testing datasets are considered in-distribution datasets, and which are out-of-distribution datasets? It seems not clear from the paper.
2.	After reading Section 5.1, I’m still confused about how to interpret the results in Figure 4. I only see as the number of components or the number of clusters increase, the simplified model becomes more similar to the original model (the JSD between the attentions decreases, while the ratio for predicting the same token increases). However, I’m not sure how one can conclude that there is a generalization gap between the simplified model and the original model. I’m also not sure how one can conclude that the simplified model underestimate/overestimate the generalization ability of the original model. Is it more appropriate to compare the prediction accuracy of the simplified model with that of the original model on both in-distribution and out-of-distribution data?

Furthermore, combined with Question 1, if the testing set named **IID** is considered in-distribution, and the testing set named **Unseen struct (len>32)** is considered out-of-distribution, then why are the curves on these two testing sets so similar to each other?

---

> ### Author Response · Authors · 2023-11-22
> **Author response to Reviewer iDw5**
>
> Thank you for your helpful comments! We are glad you find that our paper addresses an important question about the faithfulness of explanations. And thank you for pointing out areas where the presentation of the paper can be improved. We respond to your questions below and have incorporated your suggestions into our updated draft.
>
> > Introducing the Dyck languages
>
> Thank you for the suggestion! We have revised section 2.1 and Appendix A.1 to present the Dyck languages more clearly. In particular, please see Table 1 in the appendix, which provides toy examples illustrating the different generalization sets.
>
> > The phrase “simplified models” can be misleading
>
> Thank you for pointing this out. We agree that the phrase “simplified models” could be more precise. In this paper, we simplify the representations of a model, using tools like dimensionality reduction, clustering, and discretization, and then analyze these simplified representations. In other words, we are essentially replacing the original model with a simplified proxy model which uses a more limited—and thus easier to interpret—set of features. We have revised our draft to communicate this more precisely in the introduction.
>
> > Which of the testing datasets is considered in-distribution / IID vs. Unseen struct
>
> We apologize for any confusion in our presentation of the different testing splits. We constructed these splits to test how the model generalized to different kinds of unseen data. In each split, some aspect of the data can be considered “out-of-distribution” and some can be considered “in-distribution”–for example, in the “Unseen structure” datasets, the structures were not seen during training, but the nesting depth was seen.
>
> Regarding the IID split: The subset was constructed by sampling from the same distribution used to construct the training set but rejecting any strings that appeared in the training set. It turns out that almost all sequences in this subset have unseen structures. (The number of bracket structures at length $n$ is given by the n-th Catalan number, so the chance of sampling the same bracket structure twice is very low.) We have updated Section 2.1 to clarify the sampling strategy of different generalization sets. We have updated Figures 1-3 to focus on generalization splits more clearly. Additionally, we have added more details to the Appendix A.1 to further clarify the IID split, and positioning of our experiments with respect to the prior work by Murty et al. [1], who introduced the structural generalization split we adopt here.
>
> [1] Murty et al., 2023. Grokking of Hierarchical Structure in Vanilla Transformers.
>
> > How to interpret the results in Figure 4. How can one conclude that there is a generalization gap between the simplified model and the original model? How can one conclude that the simplified model underestimate/overestimate the generalization ability of the original model? Is it more appropriate to compare the prediction accuracy of the simplified model with that of the original model on both in-distribution and out-of-distribution data?
>
> In Figure 4 (Figure 2 in the revision), we concluded that there is a generalization gap because the approximation quality metrics are better on in-distribution data (e.g. comparing seen structures vs. unseen structures, or seen depths vs. unseen depths). This implies that the simplified models are better approximations of the original model on the in-distribution data than the out-of-distribution data. We concluded that the simplified model underestimates generalization because the original model gets near-perfect accuracy on all generalization splits (with the exception of depth generalization). Therefore, the fact that the simplified model makes a different prediction than the original model implies that it makes the wrong prediction, which would lead to an underestimate of the generalization ability of the original model. To conclude that one-hot attention overestimates generalization, we do compare the prediction accuracy in Figure 5b (Figure 3b in the revision): the one-hot attention model achieves higher accuracy on the depth generalization split than the original model.

---

> > ### Comment · Reviewer_iDw5 · 2023-11-23
> >
> > I'd like to thank the authors for their responses. I appreciate the added introduction of the Dyck language, and the experiment on code completion. There are still some concerns remain:
> >
> > 1. I'm still confused about how one would consider a testing set *in-distribution* or *out-of-distribution*. I think one important claim of the paper is that "simplified models" (models obtained from dimensionality reduction or clustering) may well capture the original model's behavior on in-distribution data, but fail to do so on out-of-distribution data. To validate this claim, a clear definition (or a dichotomy) of in-distribution data and out-of-distribution data is needed. To this end, some parts of the response seem confusing to me:
> > -  the claim "some aspect of the data can be considered 'out-of-distribution' and some can be considered 'in-distribution' "makes the boundary between *in-distribution* or *out-of-distribution* testing sets even more ambiguous.
> >
> > - it seems like there is actually **no** in-distribution testing set in the experiments. From the response, the testing set named **IID** is also considered out-of-distribution, correct? Then, does it mean that only the training set is considered *in-distribution*, but all other testing sets are considered *out-of-distribution*? If so, I think it is more interesting to consider **both an in-distribution testing set and an out-of-distribution testing set**. I'd like to give an example of image classification. For example, let us consider a training set in which most dog images have a background of grass. Then, an in-distribution testing set also consists of other dog images (not seen in the training set) with the grass background, while an out-of-distribution testing set consists of dog images with a background of water. This can be a more interesting setting.
> >
> > 2. About the validation of the claim that "the simplified model underestimate/overestimate the generalization ability of the original model". In the response, the authors seem to make the following explanation: because the original model achieves near-perfect accuracy on out-of-distribution testing sets, and the output of the simplified model deviates heavily from that of the original model on these testing sets, one can conclude that the simplified model does not perform well on these testing sets, which means they "underestimate" the generalization ability. However, this chain of logic is quite indirect and awkward. I would suggest conducting a direct experiment to validate this claim.
> >
> > I'd like to keep my current score, and I hope the paper can be continuously improved in the future.

---

### Author Response · Authors · 2023-11-22
**General response**

Thank you to the reviewers for their helpful comments! We are glad that the reviewers find that our paper addresses an important problem, the experimental analysis is detailed, the results are interesting, and the paper is well-written. The reviewers have also pointed out some areas where the paper could be improved, and we have uploaded a revised version of our draft incorporating these suggestions. Our updates are written in red text.

In particular, several of the reviewers noted that the contribution of the paper might be limited by our focus on the Dyck languages. We originally chose to focus on this setting because it captures important aspects of natural languages, admits a variety of systematic generalization splits, yet is simple enough to allow us to gain some insight into how simplifying representations affects the model on a mechanistic level. However, we have conducted a number of additional experiments to investigate whether our findings extend to larger models trained on a more naturalistic setting: predicting the next character in a dataset of computer code. Code completion is a common application for large language models, and it is a natural next step from the Dyck setting because it involves both “algorithmic” reasoning (including bracket matching) and more naturalistic language modeling. These experiments are described in Section 5 in the revision. (To stay within the space constraint, we moved some discussion of the Dyck models to section B.1 in the supplement.)

Specifically, we trained character-level language models on Java functions and evaluated generalization to functions with deeper nesting depths, and functions in other programming languages. These experiments also revealed a consistent generalization gap: simplified proxy models are more faithful to the original model on in-distribution data and worse on OOD data. Next, we broke down these results according to different subsets of the code completion task, and found that the generalization gap is notably larger on some types of predictions. In particular, the generalization gap is larger on subtasks that can be seen as more “algorithmic”, including predicting closing brackets and copying variable names from earlier in the sequence. The gap is smaller when predicting whitespace characters and keywords, perhaps because these predictions rely more on local context. We were able to attribute some of these generalization gaps to an “induction head” mechanism, which copies words from earlier in the sequence; simplifying this head leads to large gaps in approximation quality between in-domain and out-of-domain samples, meaning that the low-dimensional approximation underestimates the extent to which the induction head mechanism will generalize to unseen distributions.

Overall, these experiments provide evidence that generalization gaps also occur in naturalistic settings, and they provide additional insight into how these gaps are related to different properties of sequence modeling tasks. Specifically, generalization gaps might be most pronounced in “algorithmic” settings where the model must use a particular feature in a precise, context-dependent way, and there are fewer or no other proxies of that feature available in the data. The effect is diminished in settings where a variety of local features contribute to the model’s prediction. We hope that these additional experiments help to address concerns about the scope of our analysis, and we hope the reviewers will consider reevaluating their scores.

---

### Meta-Review · Area_Chair_Ub6s · 2023-12-06

**Metareview:**

This paper studies deep neural network models that are being simplified using techniques such as dimensionality reduction. It finds that even though these simplified models may agree well with the original model on the training data, this is often not the case on OOD data. After an active discussion between authors and reviewers, the paper remains very borderline, with three reviewers leaning towards acceptance and two towards rejection. While the reviewers praised the importance of the problem, the interestingness of the result, and the clarity of the writing, they were critical of the narrow empirical setting. The authors have extended their experiments with a second setting (code generation) during the rebuttal, but the reviewers overall still feel like the paper could benefit from adding a larger swath of different settings (e.g., common benchmark problems) and would then need to undergo another round of review to ensure that the claims are still supported by the new evidence. I believe that this could be a really interesting contribution and would encourage the authors to take the reviewer feedback seriously and resubmit a revised version of the paper in the future.

**Justification For Why Not Higher Score:**

the reviewers believe that the paper could be significantly improved for a resubmission

**Justification For Why Not Lower Score:**

N/A

---

### Decision · Program_Chairs · 2024-01-16

Reject